# UNC-6/Netrin promotes both adhesion and directed growth within a single axon

Ev L Nichols[1], Joo Lee[1†], Kang Shen[1,2,3]*

[1]Department of Biology, Stanford University, Stanford, United States; [2]Howard Hughes Medical Institute, Stanford University, Stanford, United States; [3]Wu Tsai Neurosciences Institute, Stanford University, Stanford, United States

## eLife Assessment

These studies make a **fundamental** contribution to our understanding of axon-guidance mechanisms, focusing on the role of UNC-6/Netrin in the long-range growth and targeting of axons. Using state-of-the-art genetics and in vivo imaging, the authors provide **compelling** support for the finding that UNC-6/Netrin can act via both chemotaxis and haptotaxis. This work will be of interest to a wide variety of cell and developmental biologists and neuroscientists.

*For correspondence:
kangshen@stanford.edu

Present address: †Department of Developmental Biology, Stanford University, Stanford, United States

Competing interest: The authors declare that no competing interests exist.

**Abstract** During development axons undergo long-distance migrations as instructed by guidance molecules and their receptors, such as UNC-6/Netrin and UNC-40/DCC. Guidance cues act through long-range diffusive gradients (chemotaxis) or local adhesion (haptotaxis). However, how these discrete modes of action guide axons in vivo is poorly understood. Using time-lapse imaging of axon guidance in *C. elegans*, we demonstrate that UNC-6 and UNC-40 are required for local adhesion to an intermediate target and subsequent directional growth. Exogenous membrane-tethered UNC-6 is sufficient to mediate adhesion but not directional growth, demonstrating the separability of haptotaxis and chemotaxis. This conclusion is further supported by the endogenous UNC-6 distribution along the axon's route. The intermediate and final targets are enriched in UNC-6 and separated by a ventrodorsal UNC-6 gradient. Continuous growth through the gradient requires UNC-40, which recruits UNC-6 to the growth cone tip. Overall, these data suggest that UNC-6 stimulates stepwise haptotaxis and chemotaxis in vivo.

## Introduction

Axon guidance is critical for establishing functional neural circuits. Given the importance of axon growth in laying the morphological foundations for neural circuitry, axon guidance has been of interest to neuroscientists since Ramón y Cajal first observed growing axons (termed growth cones) in histological examinations of the embryonic nervous system (*Ramón y Cajal, 1911*). Early cell ablation studies established that growing axons rely on anatomical landmarks, or intermediate targets, to properly navigate through their environment. A classic example of intermediate targets is in the grasshopper hindlimb, where pioneering Ti1 sensory axons navigate from the tip of the limb to the animal's central nervous system. Instead of following the shortest route, Ti1 axons grow toward a series of guidepost cells whereby they sequentially readjust their trajectory upon reaching a guidepost (*Keshishian and Bentley, 1983*). In fact, these intermediate cells are necessary for proper sensory axon guidance as their ablation results in Ti1 axon misrouting (*Bentley and Caudy, 1983b*; *Bentley and Caudy, 1983a*). Studies in other model organisms have identified additional examples of intermediate targets across phylogeny: the ventral nerve cord (VNC) in *C. elegans* (*Hedgecock et al., 1990*), the *Drosophila* midline (*Klämbt et al., 1991*), and the floor plate (*Bernhardt et al., 1992*; *Kennedy et al.,*

*1994*; *Serafini et al., 1994*) and optic chiasm in vertebrates (*Chung et al., 2000*). However, given the transient nature of a growth cone's interactions with its intermediate targets, detailed in vivo, cell biological studies of these interactions remain sparse.

The stereotyped axonal behaviors at the intermediate target require molecular cues which confer directional information to growing axons. A critical advance of the field was the identification of numerous ligands (Netrins, Slits, Semaphorins, and Ephrins) responsible for axon guidance and their cognate receptors (*Kolodkin and Tessier-Lavigne, 2011*). These signaling pathways are generally characterized as either attractive or repulsive, depending on the receptor–ligand combination (*Comer et al., 2019*). For example, Netrin attracts axons when signaling through a receptor homodimer of Deleted in Colorectal Cancer (DCC) or Neogenin1 (Neo1) whereas it repels axons when bound to a receptor dimer that includes UNC-5 (*Kolodkin and Tessier-Lavigne, 2011*).

Netrin was initially proposed to attract axons as a chemotactic cue mediated by a long-range action of diffusible Netrin present in a gradient (*Comer et al., 2019*). Initial characterizations of Netrin in the vertebrate spinal cord identified it as a diffusive cue produced by the floor plate through spinal explants (*Kennedy et al., 1994*; *Serafini et al., 1994*). More recently, in vivo examinations have visualized a Netrin-1 gradient emanating from the floor plate (*Dominici et al., 2017*; *Kennedy et al., 2006*). Together, these data support a role for Netrin in stimulating chemotaxis.

Alternatively, Netrin can stimulate haptotactic migration through local, contact-dependent interactions between the growth cone and an anchored Netrin ligand (*Comer et al., 2019*). Haptotactic mechanisms of Netrin guidance are supported by experiments in the *Drosophila* nervous system where Netrin is required for commissural axons to cross the midline. Artificial tethering of Netrin to the cell membrane at the midline is sufficient to drive commissural axon orientation and midline crossing (*Brankatschk and Dickson, 2006*). Similarly, in the *Drosophila* visual system, R8 photoreceptors utilize local Netrin and Frazzled (Fra)/DCC to stabilize their axons at the neuropil (*Timofeev et al., 2012*). Mutants for *net*, *fra*, or *trim9* (a downstream regulator of Netrin signaling) navigate normally to the neuropil but fail to stabilize, eventually retracting (*Akin and Zipursky, 2016*). Notably, evidence of Netrin-induced haptotaxis is not limited to *Drosophila*. Investigations in the vertebrate spinal cord and rhombic lip have identified local roles for Netrin in promoting axon growth and confinement within the central nervous system (*Varadarajan et al., 2017*; *Yung et al., 2018*).

In *C. elegans*, current understandings posit that the VNC produces the Netrin homolog UNC-6 which forms a ventrodorsal gradient (*Wadsworth et al., 1996*). Axons navigating ventrally are proposed to sense this UNC-6/Netrin gradient and chemotax toward the UNC-6 source at the VNC. Here, we utilize time-lapse imaging in *C. elegans* to visualize growth cone behaviors during ventral axon guidance in an intact nervous system. These experiments identify the ventral sublateral region of the worm as a common intermediate target where ventral growth cones stabilize before further migration. Both stabilization and growth depend on UNC-6/Netrin and UNC-40/DCC. Non-diffusible UNC-6 is sufficient to rescue axon stabilization but not growth. Last, imaging of endogenous UNC-6 localization during axon outgrowth is consistent with a model where the extending growth cone enriches UNC-6 around the cell membrane. Together, these experiments support a model where haptotaxis and chemotaxis act successively in UNC-6-mediated axonal migration.

## Results

### The ventral sublateral region is an intermediate target for ventral axons

*C. elegans* is a canonical model for axon guidance given its conservation of axon guidance genes and stereotyped neuronal identities and morphologies (*Chisholm et al., 2016*). While the majority of *C. elegans* neurons are born embryonically, a subset is born during larval development which enables direct observations of axon guidance by time-lapse imaging experiments. Of these postembryonic neurons, five (AVM, HSN, PVM, PDE, and PQR) initiate pioneering axons toward the ventral region of the animal where they join the VNC (*Figure 1A*). (Another neuron, PVD, is born post-embryonically but its axon follows along the axon of PDE which initiates first.) We sought to leverage this common guidance trajectory and accessible visualization to provide new insights into mechanisms of axon guidance. First, we developed a paradigm that permits complete visualization of PDE axon's ventral growth in wild-type animals. This strategy used a *Plin-32::mCherry-PH* integrated transgene to label the PDE

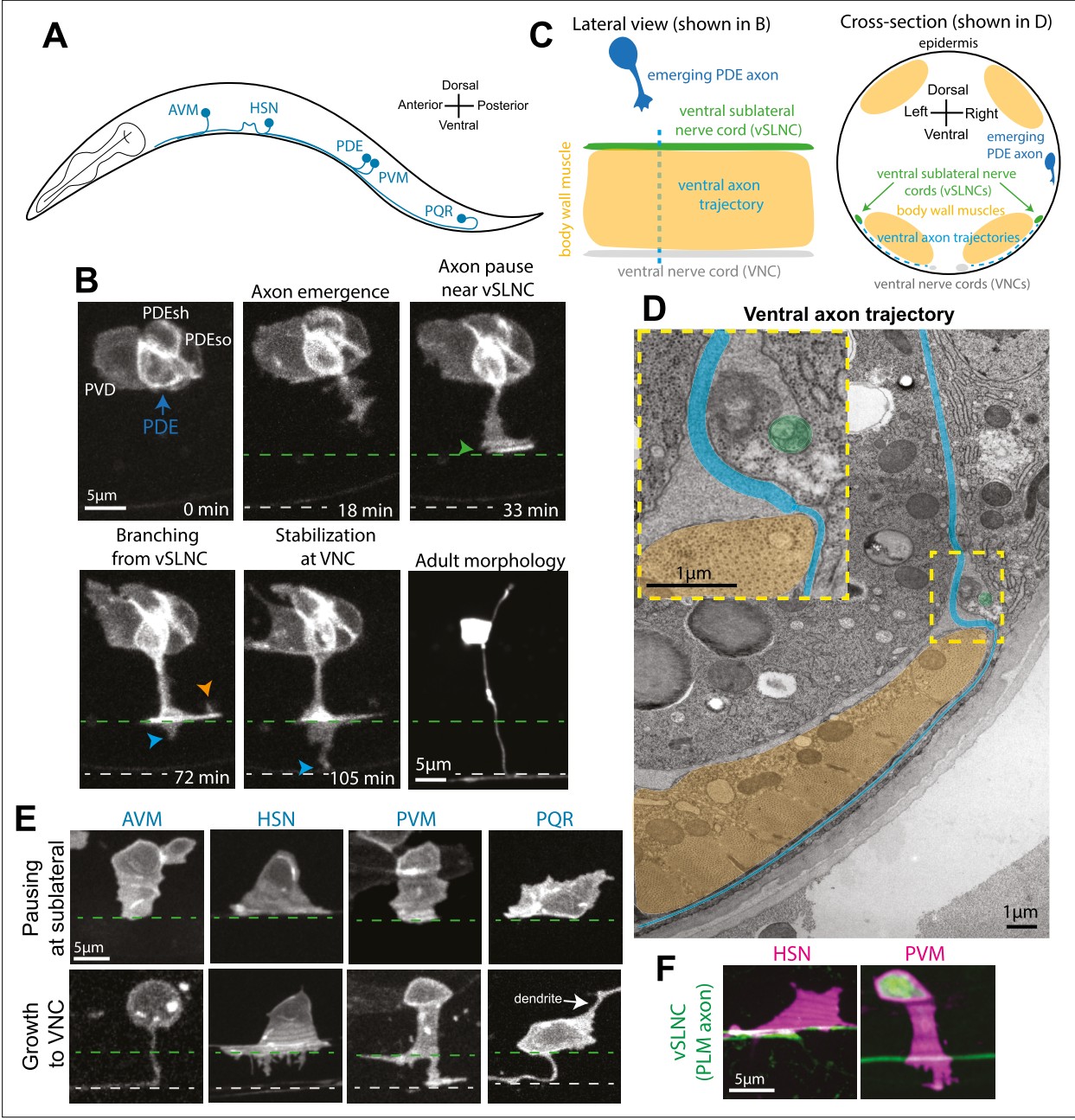

**Figure 1.** The ventral sublateral nerve cord (vSLNC) is an intermediate target for ventrally growing axons. (**A**) Diagram of *C. elegans* illustrating the anteroposterior locations of the neurons which extend ventral axons in the larval stages: AVM, HSN, PDE, PVM, and PQR. (**B**) Confocal time-lapse images of PDE axon guidance. Images were taken every 3 min for 4 hr in the L2 stage. The final image is a confocal still image visualizing the mature PDE morphology in the L4 stage. Blue arrow denotes PDE soma. White text denotes PVD neuron, PDE sheath glia (PDEsh), and PDE socket glia (PDEso). Green arrowhead denotes the axonal pause in the sublateral region. Orange arrowhead indicates a dorsal branch, and blue arrowhead denotes a ventral branch. (**C**) Cartoons showing lateral and cross-sectional views of ventral axon navigation in *C. elegans*. Ventral axons travel along the epidermal edge until they reach the sublateral region. Further ventral growth requires navigation between the epidermis and ventral body wall muscles. (**D**) Electron microscopic image of PDE axon navigation path (blue) in an adult animal. The dashed yellow box denotes the sublateral region where the axonal trajectory changes. This region is enlarged as an inset on the top left. Orange area denotes body wall muscle. Green circle denotes the vSLNC which is ensheathed by the epidermis in late larval stages. (**E**) Confocal images of AVM, HSN, PVM, and PQR axon emergence. The top row visualizes axonal contact with the vSLNC. The bottom row visualizes branching toward the ventral nerve cord (VNC). Dashed green line denotes the vSLNC and the dashed gray line denotes the VNC. White arrow denotes PQR dendrite. (**F**) Confocal images of HSN and PVM axon emergence interacting with the vSLNC. Scale bar is 5 µm in (**B, E, F**) and 1 µm in (**D**).

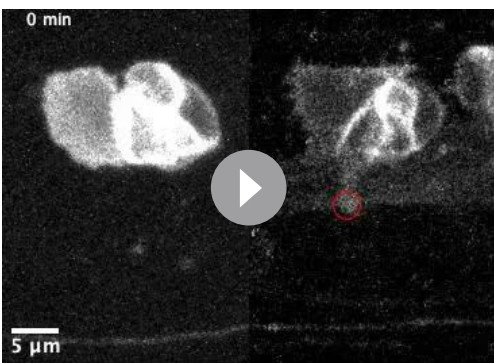

**Video 1.** Comparison of PDE axon guidance in wild-type and *unc-6(ev400)* mutants.
https://elifesciences.org/articles/100424/figures#video1

cell membrane and imaged PDE axon outgrowth every 3 min for 4 hr (*Figure 1B*, *Video 1*). The mature axonal morphology of PDE is unbranched along the dorsoventral axis before bifurcating at the VNC (*Figure 1B*). The continuous, unbranched morphology of the mature PDE axon suggests continuous ventral growth after axon initiation. Surprisingly, however, we observed that the PDE axon consistently halts its ventral growth upon reaching the region near the ventral sublateral nerve cord (vSLNC), a small axon tract that runs along the anteroposterior body axis of the animal. During its pause, the axon establishes transient collateral branch from which additional dorsal and ventral branches emerge. Eventually, one ventral branch stabilizes at the VNC, bifurcates, and initiates fasciculated growth along VNC axons. Following VNC stabilization, the transient collateral branch at the vSLNC is pruned, resulting in the unbranched PDE axonal morphology observed in adulthood. The stereotyped PDE axonal pause and consistent transient branching at vSLNC indicate that this is an intermediate guidance target.

The vSLNC sits atop the ventral body wall muscles of the worm and adjacent to the epidermis, thereby demarcating an anatomical region that separates two distinct environments through which the axon grows (*Figure 1C, D*). In adult animals, the vSLNC is ensheathed by the epidermis, but the ensheathment process begins during larval stages and is not yet completed by PDE axonogenesis (*Chalfie and Sulston, 1981*). Prior to reaching the vSLNC, the PDE axon grows along the epidermal basal lamina, but additional ventral growth past the sublateral region requires that axons wedge themselves between fibrous adhesive junctions between the body wall muscles and the epidermis (*Figure 1C, D*; *Cox and Hardin, 2004*). Given these distinct growth environments, the sublateral region (either the vSLNC itself or the body wall muscles) serves as an intermediate target during ventral axon navigation. This stark contrast in growth environments correlates with alternate forms of morphological axon growth. Prior to reaching the vSLNC, the growth cone of PDE is wide and reminiscent of a lamellipodium. However, ventral axon branching from the transient collateral branch are consistent with filopodia-based motility.

Given that the body wall muscles tile the anteroposterior axis of the entire animal, other ventrally directed axons also must undergo a migration through these distinct regions. We next hypothesized that the other ventrally growing axons would similarly utilize the ventral sublateral region as an intermediate target. To test this, we imaged axon emergence in AVM, HSN, PVM, and PQR neurons. While these neurons position their somas more ventrally, we observe similar axonal trajectories as the PDE axon: axons initially emerge ventrally as lamellipodia, then adhere upon reaching the vSLNC. Multiple ventrally directed filopodia form and a ventral branch eventually stabilizes at the axons' ultimate ventral target, the VNC (*Figure 1E*). To gauge the location of axon pausing relative to the vSLNC, we imaged HSN and PVM during axon emergence with a reporter construct expressing GFP under the control of the *mec-7* promoter region (*Figure 1F*). This reporter construct labels mechanosensory neurons, including the PLM neuron whose axon comprises the vSLNC. Taken together, these data reveal that ventral axon guidance in *C. elegans* is a stepwise process facilitated by an intermediate target near the vSLNC. Notably, dorsally migrating axons in *C. elegans* also stall their growth cones at the borders of longitudinal nerve tracks and muscle borders, forming anvil-shaped structures (*Knobel et al., 1999*). These results argue that the stepwise migration and pausing of growth cone at specific tissue structures is a general phenomenon in *C. elegans*, prompting us to understand the molecular bases of this cellular behavior.

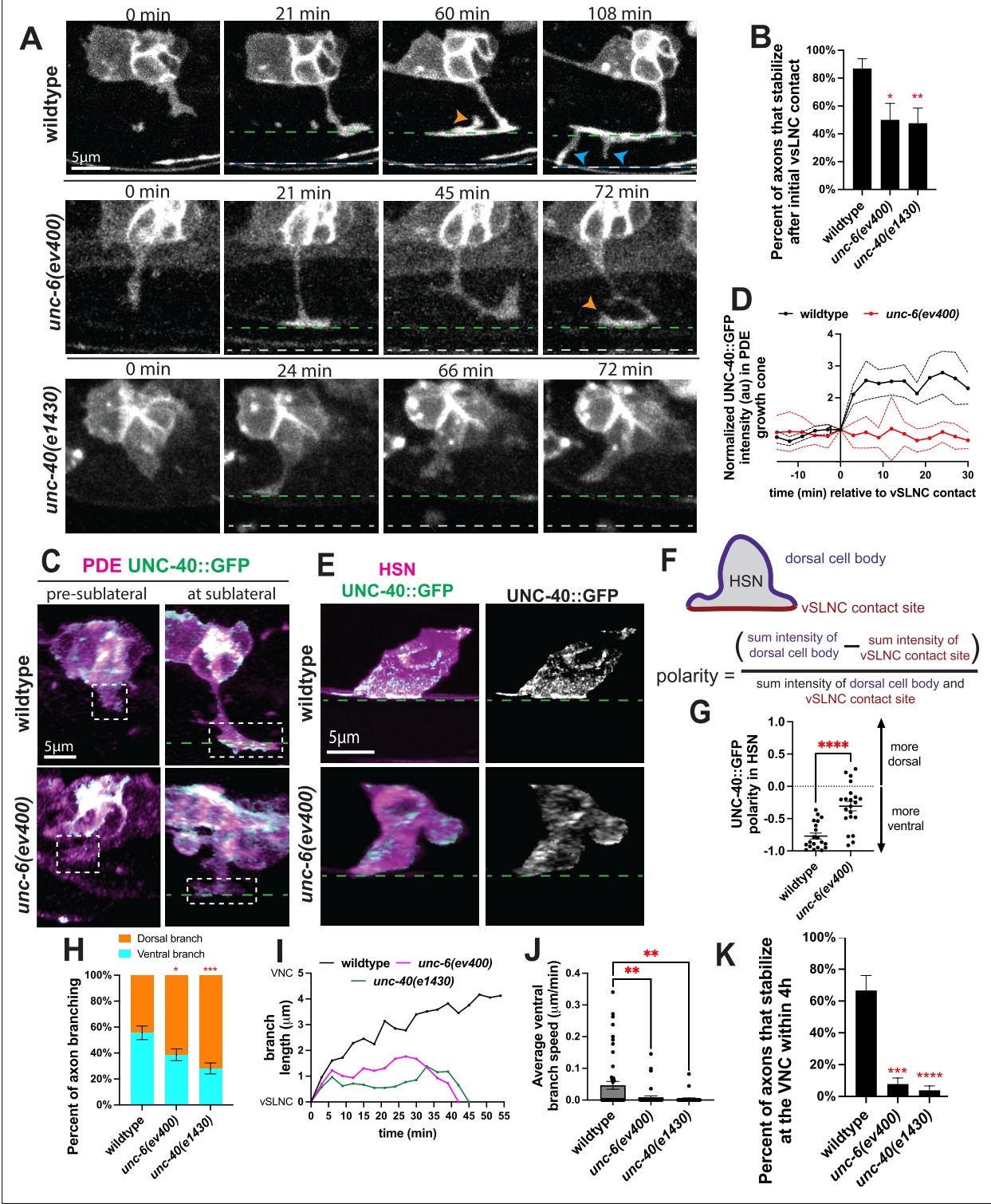

**Figure 2.** UNC-6 and UNC-40 promote stabilization and directed growth at the sublateral region. (**A**) Confocal time-lapse images of PDE axons guidance in wild-type, *unc-6(ev400)*, and *unc-40(e1430)* animals. Images were taken every 3 min for 4 hr in the L2 stage. Orange arrowhead indicates a dorsal branch, and blue arrowhead denotes a ventral branch. Dashed green line denotes the ventral sublateral nerve cord (vSLNC) and the dashed gray line denotes the ventral nerve cord (VNC). (**B**) Percentage of axons that stabilize in the sublateral region upon initial contact in wild-type (*n* = 23), *unc-6(ev400)* (*n* = 18), and *unc-40(e1430)* (*n* = 21) animals. SE is shown. (**C**) Airyscan images of PDE axon navigation and endogenous UNC-40::GFP growth cone localization in wild-type and *unc-6(ev400)* animals. Dashed white box denotes growth cone. (**D**) Normalized intensity traces of growth cone UNC-40::GFP during axon navigation normalized to sublateral contact (*t* = 0). Black line denotes wild-type animals (*n* = 8), and red line denotes *unc-6(ev400)*

*Figure 2 continued on next page*

*Figure 2 continued*

animals (*n* = 8). SEM is shown. (**E**) Airyscan images of HSN axon emergence and endogenous UNC-40::GFP localization in wild-type and *unc-6(ev400)* animals. (**F**) Cartoon schematic of polarity measurements and calculation in (**G**). (**G**) UNC-40::GFP polarity in HSN in wild-type (*n* = 20) and *unc-6(ev400)* (*n* = 21) animals. Positive values denote dorsal polarization, and negative values denote ventral polarization. SEM is shown (**H**) Proportion of branches formed dorsally (orange) and ventrally (cyan) from the sublateral region in wild-type (*n* = 90 branches, 24 animals), *unc-6(ev400)* (*n* = 114 branches, 25 animals), and *unc-40(e1430)* (*n* = 114 branches, 28 animals) animals. SE is shown. (**I**) Representative traces of ventral branch extension in wild-type (black), *unc-6(ev400)* (magenta), and *unc-40(e1430)* (green) animals. (**J**) Average ventral branch extension speed in wild-type (*n* = 50 branches, 24 animals), *unc-6(ev400)* (*n* = 43 branches, 25 animals), and *unc-40(e1430)* (*n* = 32 branches, 28 animals) animals. SEM is shown. (**K**) Percent of axons that stabilize that the VNC within the 4 hr imaging window in wild-type (*n* = 24), *unc-6(ev400)* (*n* = 25), and *unc-40(e1430)* (*n* = 28) animals. SE is shown. Scale bars denote 5 µm in (**A, C, E**). Fisher's exact test with Bonferonni's correction is used in (**B, H, K**). Unpaired Student's *t*-test is used in (**G**). Ordinary one-way ANOVA with multiple comparisons is used in (**J**). * denotes $p < 0.05$, ** denotes $p < 0.01$, *** denotes $p < 0.001$, **** denotes $p < 0.0001$. All comparisons to wild-type unless otherwise noted.

The online version of this article includes the following source data and figure supplement(s) for figure 2:

**Source data 1.** Quantification for *Figure 2*, *Figure 2—figure supplement 1*.

**Figure supplement 1.** Axon emergence and UNC-40 polarization are dependent on UNC-6.

## UNC-6/Netrin signaling is required for multiple steps of ventral axon guidance

Circumferential axon migration in *C. elegans* is primarily driven by signaling through UNC-6/Netrin and its receptors (*Hedgecock et al., 1990*; *Wadsworth et al., 1996*). Dorsal guidance requires UNC-6 to signal through both its receptors, UNC-5 and UNC-40/DCC. Ventral guidance does not require UNC-5 signaling but is dependent on UNC-40 (*Hedgecock et al., 1990*). Most evidence supporting this notion is based on analyses of post-developmental axonal morphologies in mutants instead of visualizing the developmental process. We used time-lapse imaging of PDE axon outgrowth in null mutants of *unc-6* (*ev400*) and *unc-40* (*e1430*) to understand whether UNC-6/UNC-40 signaling is required for one or all of the events: (1) initial axon outgrowth direction, (2) stabilization at the intermediate target, (3) directional branching from the sublateral region, or (4) ventral growth from intermediate target to VNC. Importantly, the guidance of the vSLNC axons does not rely on UNC-6 or UNC-40, so the putative intermediate target is intact in these mutants (*Li et al., 2008*). The resulting movies demonstrate that each step of ventral growth is disrupted in *unc-6* and *unc-40* mutants (*Figure 2A*, *Video 1* and *Video 2*, *Figure 2—figure supplement 1*). First, in most wild-type animals, the axon emerges from the ventral side of PDE, while axons emerge from non-ventral directions of PDE in *unc-6* and *unc-40* mutants (*Figure 2—figure supplement 1A, B*). To compare axon emergence between genotypes, we calculated the sine of angle of axon emergence which indicates where along the dorsal–ventral axis the axon is emerging. A sine value of 1 denotes dorsal emergence at 90°, 0 indicates emergence along the anterior–posterior axis, and −1 denotes 90° ventral emergence. These comparisons further implicate UNC-6 and UNC-40 in determining the direction of PDE axon emergence.

Second, over 80% of wild-type axons form a stable collateral branch immediately upon reaching the vSLNC. In contrast, nearly half of the *unc-6* and *unc-40* mutant axons form collateral branches at vSLNC but subsequently retract (*Figure 2A, B*). Typically, these axons 'probe' the vSLNC again after retraction and establish contacts with vSLNC again during the imaging window. However, retraction upon initial sublateral contact is rarely observed in wild-type animals, indicating that UNC-6 signaling stabilizes the growth cone at the sublateral intermediate target.

To further understand how UNC-6 and UNC-40 regulate axon development, we inserted a GFP

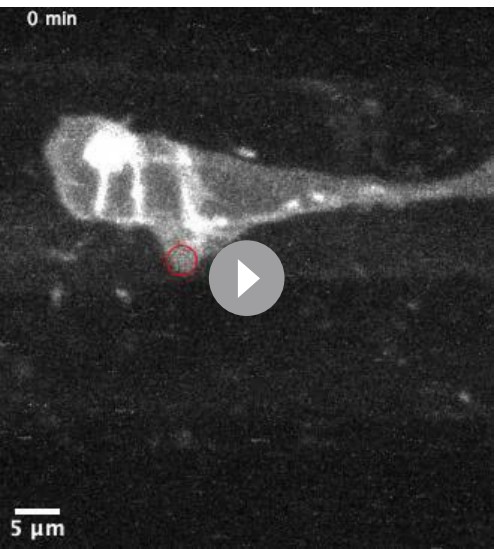

**Video 2.** PDE axon guidance in *unc-40(e1430)* mutants.
https://elifesciences.org/articles/100424/figures#video2

**Table 1.** Genetic analysis of PDE axon morphology in adulthood.

'% of axons at VNC' denotes percentage of PDE neurons with axons that extend to the ventral nerve cord (VNC). '% of axons not at VNC' denotes percentage of PDE neurons with axons that do not extend to the VNC. *N* indicates number of animals scored. p-values generated from Fisher's exact test.

| Genotype | % of axons at VNC | % of axons not at VNC | Standard error (%) | *n* | Comparison with wild-type (p) | Comparison with *unc-6(ev400)* (p) |
|---|---|---|---|---|---|---|
| Wild-type | 100.0 | 0.0 | 0.0 | 118 | | <0.0001 |
| *unc-6(ev400)* | 32.1 | 67.9 | 5.1 | 84 | <0.0001 | |
| *unc-40(e1430)* | 33.0 | 67.0 | 4.8 | 97 | <0.0001 | >0.9999 |
| *unc-6(ev400); Pmec-7::unc-6* | 31.6 | 68.4 | 4.8 | 95 | <0.0001 | >0.9999 |
| *unc-6(ev400); Punc-4::unc-6* | 35.2 | 64.8 | 5.1 | 88 | <0.0001 | 0.7475 |
| *unc-6(ev400); Pmec-7::unc-6; Punc-4::unc-6* | 68.6 | 31.4 | 5.0 | 86 | <0.0001 | <0.0001 |
| *Pmec-7::unc-6* | 100.0 | 0.0 | 0.0 | 127 | >0.9999 | <0.0001 |
| *Punc-4::unc-6* | 100.0 | 0.0 | 0.0 | 102 | >0.9999 | <0.0001 |
| *slt-1(eh15)* | 93.3 | 6.7 | 2.2 | 134 | 0.0039 | <0.0001 |
| *unc-6(ev400); slt-1(eh15)* | 22.3 | 77.7 | 3.7 | 130 | <0.0001 | 0.1148 |
| *madd-2(ok2226)* | 94.7 | 5.3 | 2.3 | 95 | 0.0166 | <0.0001 |
| *unc-6(ev400); madd-2(ok2226)* | 43.6 | 56.4 | 4.9 | 101 | <0.0001 | 0.1299 |
| *unc-40(4KR)* | 94.9 | 5.1 | 2.2 | 99 | 0.0187 | <0.0001 |
| *unc-40(4KR); madd-2(ok2226)* | 94.7 | 5.3 | 1.9 | 133 | 0.0155 | <0.0001 |

tag at the C-terminus of the endogenous *unc-40* locus. The resulting animals show UNC-40::GFP fluorescence in the nerve ring and major nerve cords of the worm (VNC and dorsal nerve cord), consistent with a published endogenous UNC-40::mNeonGreen strain (*Figure 2—figure supplement 1C–E*; *Jayadev et al., 2022*). We next analyzed movies of UNC-40::GFP during PDE axon outgrowth. These movies demonstrate that the PDE growth cone contains low levels of UNC-40::GFP prior to reaching the sublateral region. However almost immediately upon contacting the sublateral, UNC-40::GFP fills the growth cone along the transient collateral branch (*Figure 2C, D*). However, *unc-6(ev400)* mutants fail to increase growth cone UNC-40::GFP upon reaching the sublateral region and instead demonstrate seemingly random fluctuations in growth cone UNC-40::GFP intensity (*Figure 2C, D*). These results suggest that UNC-6 regulates UNC-40 levels in the growth cone.

Given similar morphological polarization PDE and HSN growth cones, we sought to generalize these findings in HSN which develops more slowly than PDE and is therefore more amenable to imaging axon engagement with the ventral sublateral intermediate target. Indeed, endogenous UNC-40::GFP is enriched at the ventral surface of HSN cell body where its axon emerges. However, this enrichment is absent in *unc-6* mutants, consistent with previous studies using overexpressed UNC-40::GFP constructs (*Figure 2E*; *Adler et al., 2006*). To quantify UNC-40 polarity, we measured the total GFP intensity along the periphery of the dorsal soma and sublateral contact site of HSN. The overall UNC-40::GFP polarity was calculated by dividing the difference between the two subcellular locations by the total GFP intensity along the cell periphery (*Figure 2F*, *Figure 2—figure supplement 1F, G*). The resulting polarity index ranges from –1 to 1 with negative numbers indicating a ventral UNC-40::GFP bias and positive numbers indicating a dorsal bias. A polarity value of 0 denotes an equal dorsoventral distribution of UNC-40::GFP. We also calculated the polarity of UNC-40 in AVM,

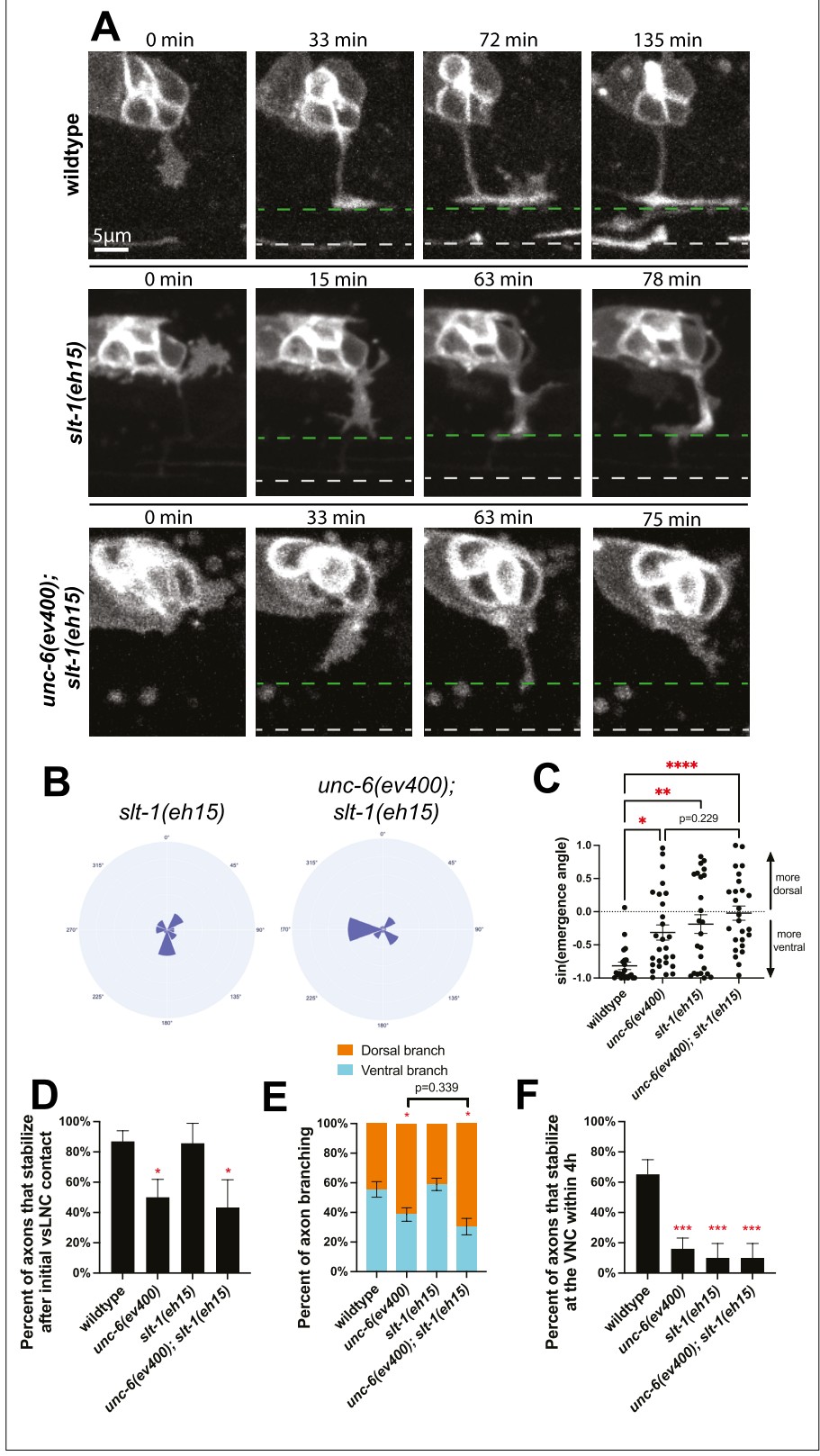

**Figure 3.** SLT-1 directs axon emergence trajectory. (**A**) Confocal time-lapse images of PDE axons guidance in wild-type, *slt-1(eh15),* and *unc-6(ev400); slt-1(eh15)* animals. Images were taken every 3 min for 4 hr in the L2 stage. Dashed green line denotes the ventral sublateral nerve cord (vSLNC) and the dashed gray line denotes the ventral nerve cord (VNC). (**B**) Rose plot histogram of PDE axon emergence in *slt-1(eh15)* (*n* = 24) and *unc-6(ev400);*

*Figure 3 continued on next page*

*Figure 3 continued*

*slt-1(eh15)* (*n* = 26) animals. (**C**) Graph comparing the sine function of axon emergence angle in wild-type (*n* = 22), *unc-6(ev400)* (*n* = 27), *slt-1(eh15)* (*n* = 24), and *unc-6(ev400); slt-1(eh15)* (*n* = 26) animals. 1 indicates 90° dorsal axon emergence and –1 indicates 90° ventral axon emergence. (**D**) Percentage of axons that stabilize in the sublateral region upon initial contact in wild-type (*n* = 23), *unc-6(ev400)* (*n* = 18), and *slt-1(eh15)* (*n* = 8), and *unc-6(ev400); slt-1(eh15)* (*n* = 7) animals. SE is shown. (**E**) Proportion of branches formed dorsally (orange) and ventrally (cyan) from the sublateral region in wild-type (*n* = 90 branches, 24 animals), *unc-6(ev400)* (*n* = 114 branches, 25 animals), and *slt-1(eh15)* (*n* = 139 branches, 10 animals), and *unc-6(ev400); slt-1(eh15)* (*n* = 69 branches, 10 animals) animals. SE is shown. (**F**) Percent of axons that stabilize that the VNC within the 4 hr imaging window in wild-type (*n* = 24), *unc-6(ev400)* (*n* = 25), and *slt-1(eh15)* (*n* = 10), and *unc-6(ev400); slt-1(eh15)* (*n* = 10) animals. SE is shown. Scale bar denote 5 µm in (**A**). Ordinary one-way ANOVA with multiple comparisons is used in (**C**). Fisher's exact test with Bonferonni's correction is used in (**D–F**). Ordinary one-way ANOVA with multiple comparisons is used in (**C**). * denotes p < 0.05, ** denotes p<0.01, *** denotes p < 0.001, **** denotes p < 0.0001. All comparisons to wild-type unless otherwise noted.

The online version of this article includes the following source data for figure 3:

**Source data 1.** Quantification for *Figure 3*.

PVM, and PQR in wild-type and *unc-6(ev400)* animals. Wild-type animals display negative polarity calculations, but in *unc-6* mutant animals, the polarity index is close to 0 indicating that UNC-40 is evenly distributed between the dorsal and ventral neuronal domains (*Figure 2E, G*; *Figure 2— figure supplement 1H–M*). Together, these data indicate that upon vSLNC contact, UNC-6 signaling enriches UNC-40 at the contact site and facilitates adhesion with the surrounding environment.

Once the axon establishes a lateral branch at the vSLNC, it generates dorsally and ventrally oriented filopodia (*Figure 2A*). These filopodia are dynamic and undergo constant extension and retraction. Using time-lapse movies to capture the dynamics, we quantified the directionality, length, growth speed and stability of these branches (*Figure 2H–K*). Wild-type animals generate slightly more ventral branches than dorsal ones (55.6% ventrally directed branches vs. 44.4% dorsal). However, *unc-6* and *unc-40* mutants have a strong reversal of this bias where 61.4% and 71.9% of branching, respectively, is directed dorsally (*Figure 2H*). These data suggest that UNC-6/UNC-40 signaling promotes ventral branching. To further test this, we analyzed the growth trajectories of individual ventral branches. This analysis demonstrates that ventral growth stalls in *unc-6* and *unc-40* mutants, manifesting as a decreased average growth speed (*Figure 2I, J*). While *unc-6* and *unc-40* mutant PDE axonal filopodia reach the VNC, very few stabilize within the imaging window. Instead, most branches retract, indicating that UNC-6 and UNC-40 are also required for stabilizing filopodia once it reaches the VNC (*Figure 2K*).

Taken together, these data indicate that UNC-6 and UNC-40 are required for several steps of axon guidance including the direction of axonal initiation, axonal adhesion at the intermediate target, directional growth toward the VNC and axonal stabilization at the VNC. While the developmental process is dynamic with numerous growth and retraction events, UNC-6 and UNC-40's impact on the stepwise events compound to ultimately disrupt the adult morphology of ventrally directed axons. In *unc-6(ev400)* mutants, 67.9% ± 5.1% of PDE axons and 66.9% ± 4.3% of HSN axons fail to reach the VNC by adulthood, compared with only 0% of PDE axons and 0.91% ± 0.9% of HSN axons in wild-type animals (*Tables 1 and 2*; *Figure 2—figure supplement 1N*).

## SLT-1/Slit orients axon emergence in parallel to UNC-6

In addition to UNC-6, SLT-1/Slit also directs ventral trajectories of *C. elegans* axons through its receptor SAX-3/Robo (*Hao et al., 2001*). SLT-1 is produced in dorsal regions of the worm and functions as a repellant. Given this, we next sought to leverage our dynamic imaging platform to analyze the precise growth cone behaviors that SLT-1 is responsible for. The resulting movies of *slt-1(eh15)* animals reveal that these PDE growth cones emerge from more dorsal regions of the cell body (*Figure 3A–C*, *Video 3*). Often, these misrouted axons corrected their trajectory by reorienting growth to the ventral region of the worm, likely in response to UNC-6. Further, we did not observe any defects in axon stabilization at the vSLNC or directional branching in *slt-1* mutants (*Figure 3D, E*). Animals lacking SLT-1 were less likely than wild-type to stabilize at the VNC within the 4 hr imaging window (*Figure 3F*). We

posit that this stabilization defect is transient because adult neuronal morphologies in these mutants present with only mild phenotypes in comparison to *unc-6* mutants (*Tables 1 and 2*).

Given that loss of *slt-1* enhances axon guidance defects in *unc-6* mutants, we next sought to characterize PDE axon guidance in this double mutant background. We found that the angle of axon emergence in the *unc-6(ev400); slt-1(eh15)* animals are completely random with no ventral preference (*Figure 3C*, *Video 4*). The double mutant also showed strong defects in vsLNC stabilization, axonal branching and stabilization at VNC (*Figure 3D–F*). This is consistent with the strong axon trajectory phenotypes found in the adult animals of this genotype (*Tables 1 and 2*).

## MADD-2/Trim9/Trim67 promotes growth cone adhesion

Our data thus far demonstrate that PDE axon outgrowth is an excellent in vivo model of Netrin-mediated guidance where precise axonal behaviors can be probed across distinct change-of-function genotypes. Therefore, we next asked whether regulators of UNC-6/Netrin signaling might only affect growth or adhesion. For this analysis, we chose the E3 ubiquitin ligase MADD-2/Trim9/Trim67 because of its conserved role in Netrin-dependent guidance and morphological polarization of the HSN emerging axon (*Akin and Zipursky, 2016*; *Hao et al., 2010*; *Plooster et al., 2017*). We captured movies of PDE axon guidance in *madd-2* null (*ok2226*) animals (*Figure 4A*, *Video 5*). These analyses revealed that *madd-2* mutants showed normal growth cone orientation at emergence (*Figure 4—figure supplement 1A, B*). However, the majority of the *madd-2* mutant axons retracted upon vSLNC contact similar to *unc-6* mutants, suggesting that MADD-2 contributes to UNC-6-mediated adhesion to the intermediate target (*Figure 4B*).

Given UNC-40's requirement for adhesion and ventral polarization, we next asked whether MADD-2 is required for the polarized subcellular localization of UNC-40. To test this, we observed endogenous UNC-40::GFP polarization within HSN, AVM, PVM, and PQR in *madd-2* mutants. These images reveal that UNC-40 loses its ventral polarity in *madd-2* mutants (*Figure 4C, D*; *Figure 4—figure supplement 1E–N*). We confirmed these results in PDE by analyzing growth cone UNC-40 levels during axon guidance. Consistent with the results in HSN, each of AVM, PVM, and PQR, the PDE growth cone showed reduced UNC-40 polarization at the vSLNC (*Figure 4—figure supplement 1C–H*). Together, these data reveal a requirement for MADD-2 to polarize UNC-40 at endogenous levels. Further, we analyzed *madd-2; unc-6* double mutants for UNC-40::GFP polarity in HSN. These animals demonstrate enhanced mispolarization of UNC-40 (*Figure 4C, D*), suggesting that UNC-6 and MADD-2 might promote the polarity of the UNC-40 receptor in parallel pathways.

Existing literature suggests that Trim9 ubiquitinates DCC and that Netrin antagonizes this activity (*Plooster et al., 2017*). To test whether

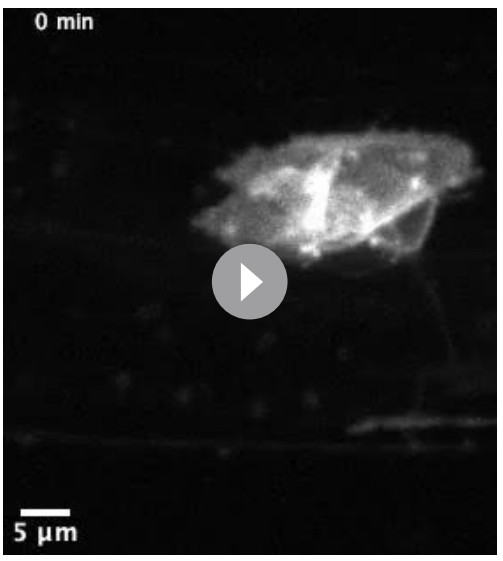

**Video 3.** PDE axon guidance in *slt-1(eh15)* mutants.
https://elifesciences.org/articles/100424/figures#video3

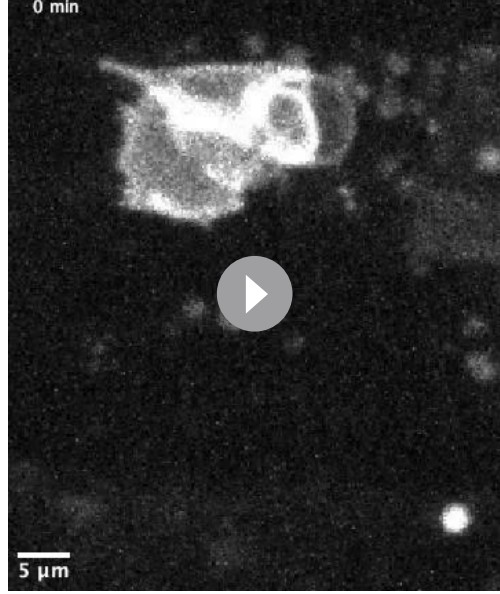

**Video 4.** PDE axon guidance in *unc-6(ev400); slt-1(eh15)* mutants.
https://elifesciences.org/articles/100424/figures#video4

**Table 2.** Genetic analysis of HSN axon morphology in adulthood.

'% of axons at VNC' denotes percentage of HSN neurons with axons that extend to the ventral nerve cord (VNC). '% of axons not at VNC' denotes percentage of HSN neurons with axons that do not extend to the VNC. *N* indicates number of animals scored. p-values generated from Fisher's exact test.

| Genotype | % of axons at VNC | % of axons not at VNC | Standard error (%) | *n* | Comparison with wild-type (p) | Comparison with *unc-6(ev400)* (p) |
|---|---|---|---|---|---|---|
| Wild-type | 99.1 | 0.91 | 0.90 | 110 | | <0.0001 |
| *unc-6(ev400)* | 33.1 | 66.9 | 4.3 | 121 | <0.0001 | |
| *unc-40(e1430)* | 30.6 | 69.4 | 4.1 | 124 | <0.0001 | 0.7840 |
| *unc-6(ev400); Pmec-7::unc-6* | 33.3 | 66.7 | 4.5 | 108 | <0.0001 | >0.9999 |
| *unc-6(ev400); Punc-4::unc-6* | 29.5 | 70.5 | 4.5 | 105 | <0.0001 | 0.6666 |
| *unc-6(ev400); Pmec-7::unc-6; Punc-4::unc-6* | 47.4 | 52.6 | 4.6 | 116 | <0.0001 | 0.0252 |
| *Pmec-7::unc-6* | 99.1 | 0.9 | 0.9 | 111 | >0.9999 | <0.0001 |
| *Punc-4::unc-6* | 100.0 | 0.0 | 0.0 | 104 | >0.9999 | <0.0001 |
| *slt-1(eh15)* | 93.6 | 6.4 | 2.3 | 110 | 0.0142 | <0.0001 |
| *unc-6(ev400); slt-1(eh15)* | 8.1 | 91.9 | 2.4 | 124 | <0.0001 | <0.0001 |
| *madd-2(ok2226)* | 92.2 | 7.8 | 2.5 | 115 | 0.0191 | <0.0001 |
| *unc-6(ev400); madd-2(ok2226)* | 44.4 | 55.6 | 4.5 | 124 | <0.0001 | 0.0881 |
| *unc-40(4KR)* | 91.9 | 8.1 | 2.4 | 124 | 0.0114 | <0.0001 |
| *unc-40(4KR); madd-2(ok2226)* | 92.2 | 7.8 | 2.5 | 116 | 0.0192 | <0.0001 |

UNC-40 ubiquitination is required for its polarization at the vSLNC, we generated *unc-40(4KR)* animals with lysine to arginine mutations at the four intracellular lysine residues adjacent to the UNC-40 transmembrane domain to hamper potential ubiquitination. These mutations were made in the UNC-40::GFP background to allow visualization of endogenous levels of mutant UNC-40. These four amino acid substitutions are sufficient to mispolarize UNC-40 in HSN, comparable to polarity measurements in *madd-2(ok2226)* animals (*Figure 4C, E, F*). Critically, UNC-40 mispolarization is not enhanced in *unc-40(4KR); madd-2(ok2226)* animals, suggesting that UNC-40 ubiquitination functions in the same pathway as MADD-2. Given that MADD-2 directly regulates UNC-40 localization through ubiquitination, these data suggest that MADD-2 ensures that UNC-40 is polarized and enriched at the growth cone at the sublateral region to promote adhesion.

In the next stage of growth, the movies of *madd-2* mutant demonstrate that PDE axons primarily initiate ventrally directed branches similar to wild-type animals, indicating that the directional growth of PDE axon is independent of MADD-2 (*Figure 4G*). This is notable given that vertebrate mutants for the MADD-2 ortholog Trim9 display aberrant branching (*Winkle et al., 2014*). However, these ventral branches also display decreased stabilization at the VNC similar to *unc-6* mutants (*Figure 4H*). Together, these experiments reveal two important insights: axonal adhesion and stabilization are dependent on MADD-2 but directional growth is not.

## Diffusible UNC-6 is dispensable for axon stabilization

UNC-6/Netrin has been proposed to function via either chemotaxis or haptotaxis (*Comer et al., 2019*; *Varadarajan et al., 2017*; *Wu et al., 2019*). Chemotactic growth occurs in response to a gradient where axons sense the Netrin concentration in their immediate environment to determine their

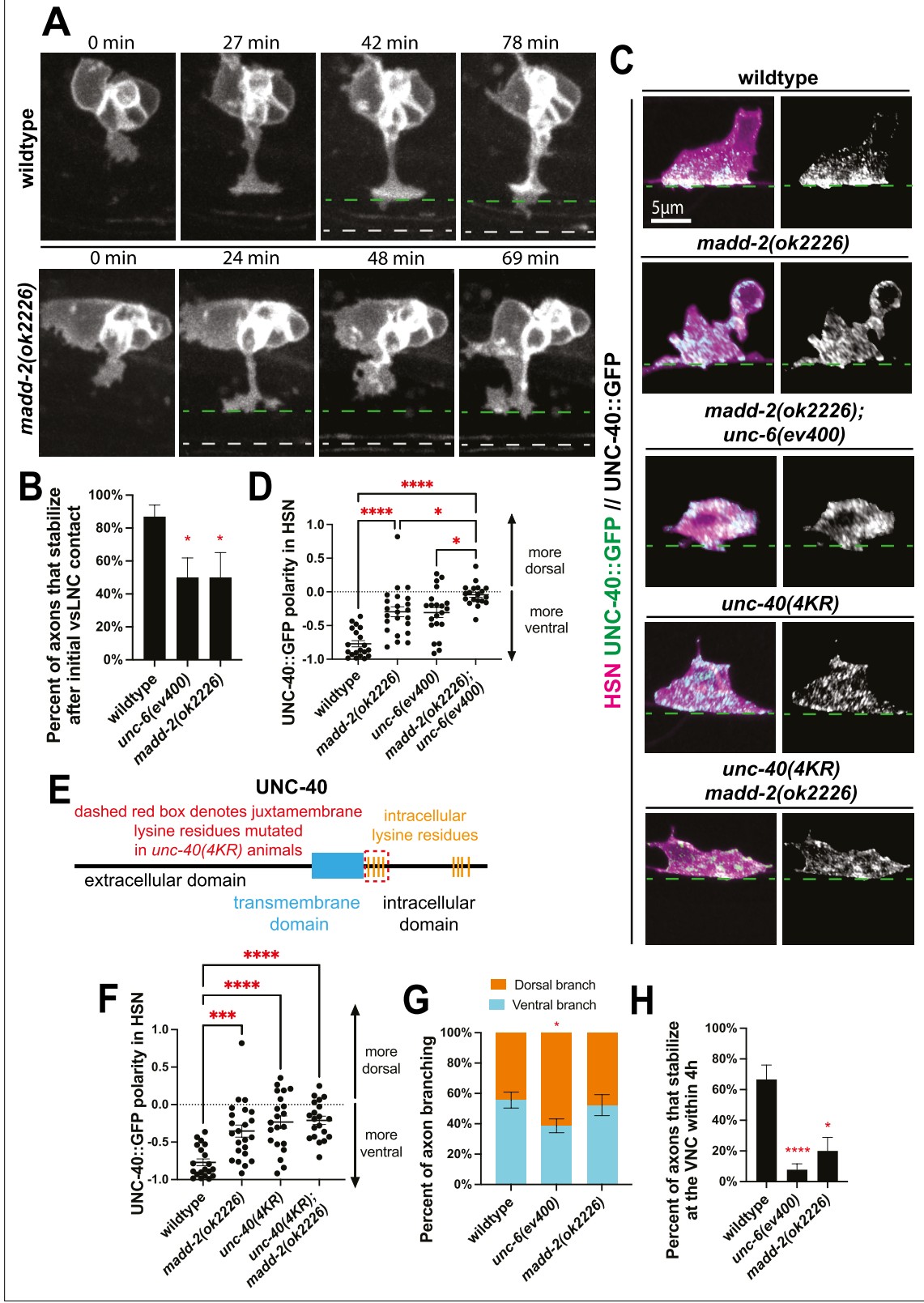

**Figure 4.** MADD-2 promotes axon stabilization but not growth. (**A**) Confocal time-lapse images of PDE axons guidance in wild-type and *madd-2(ok2226)* animals. Images were taken every 3 min for 4 hr in the L2 stage. Dashed green line denotes the ventral sublateral nerve cord (vSLNC) and the dashed gray line denotes the ventral nerve cord (VNC). (**B**) Percentage of axons that stabilize in the sublateral region upon initial contact in wild-type (*n* = 23), *unc-6(ev400)* (*n* = 18), and *madd-2(ok2226)* (*n* = 11) animals. SE is shown. (**C**) Airyscan images of HSN axon emergence and endogenous

*Figure 4 continued on next page*

*Figure 4 continued*

UNC-40::GFP localization in wild-type, *madd-2(ok2226)*, *madd-2(ok2226); unc-6(ev400)*, *unc-40(4KR)*; and *unc-40(4KR); madd-2(ok2226)* animals. (**D**) UNC-40::GFP polarity in HSN in wild-type (*n* = 20), *unc-6(ev400)* (*n* = 21), *madd-2(ok2226)* (*n* = 23), and *unc-6(ev400); madd-2(ok2226)* (*n* = 19) animals. Positive values denote dorsal polarization, and negative values denote ventral polarization. SEM is shown. (**E**) Cartoon schematic of the UNC-40 protein sequence. The dashed red box denotes the intracellular lysine residues that are mutated in *unc-40(4KR)* animals. (**F**) UNC-40::GFP polarity in HSN in wild-type (*n* = 20), *madd-2(ok2226)* (*n* = 23), *unc-40(4KR)* (*n* = 21); *unc-40(4KR); madd-2(ok2226)* (*n* = 20) animals. Positive values denote dorsal polarization, and negative values denote ventral polarization. SEM is shown. (**G**) Proportion of branches formed dorsally (orange) and ventrally (cyan) from the sublateral region in wild-type (*n* = 90 branches, 24 animals), *unc-6(ev400)* (*n* = 114 branches, 25 animals), and *madd-2(ok2226)* (*n* = 52 branches, 12 animals) animals. SE is shown. (**H**) Percent of axons that stabilize that the VNC within the 4 hr imaging window in wild-type (*n* = 24), *unc-6(ev400)* (*n* = 25), and *madd-2(ok2226)* (*n* = 12) animals. SE is shown. Scale bars denote 5 μm in (**A, C**) Fisher's exact test with Bonferonni's correction is used in (**B, G, H**). Ordinary one-way ANOVA with multiple comparisons is used in (**D, F**). * denotes $p < 0.05$, *** denotes $p < 0.001$, **** denotes $p < 0.0001$. All comparisons to wild-type unless otherwise noted.

The online version of this article includes the following source data and figure supplement(s) for figure 4:

**Source data 1.** Quantification for *Figure 4*, *Figure 4—figure supplement 1*.

**Figure supplement 1.** UNC-40 polarization, but not axon emergence, is dependent on MADD-2.

direction of travel toward or away from the Netrin source. In contrast, haptotactic growth does not rely on a long-distance gradient and instead uses local, minimally diffusive Netrin to anchor growth along a laminar surface. In the vertebrate spinal cord, commissural axons utilize both mechanisms of Netrin-mediated guidance to properly establish ventral tracts that avoid the developing motor column (*Varadarajan et al., 2017*; *Wu et al., 2019*).

Therefore, we sought to distinguish between these two UNC-6 growth mechanisms during stepwise PDE axon guidance by expressing a non-diffusible form of UNC-6 in the intermediate target region in the *unc-6* mutants. Specifically, we fused the *unc-6* coding sequence to the transmembrane domain sequence of *nlg-1/neuroligin* so that the molecule is tethered to the plasma membrane (*Teichmann and Shen, 2011*). We generated two membrane-tethered UNC-6 expression constructs: one under the control of *mec-7* promoter elements for expression along the vSLNC and one under the control of the *unc-4* promoter for expression at the VNC (*Figure 5A*). Importantly, UNC-4 expressing cholinergic motor neurons produce endogenous UNC-6 in larval stages and MEC-7 expressing mechanosensory neurons do not (*Basu et al., 2021*). However, membrane-tethered UNC-6 would be present on the axons in VNC but unable to diffuse toward the PDE cell body.

Using these constructs, we generated three different transgenic strains in the *unc-6* mutant background, a strain expressing *Pmec-7::unc-6*, a strain expressing *Punc-4::unc-6* and a strain carrying both. We validated that these rescue constructs were expressed in the proper cells by including a SNAP ligand-binding domain after the transmembrane domain. SNAP labeling confirms that these constructs are expressed along the vSLNC and VNC, consistent with the promoter fusions (*Figure 5—figure supplement 1A*). We also did not detect SNAP signals outside of the nerve tracks, indicating that the membrane-tethered UNC-6 is not cleaved and shed at significant quantity. Importantly, these transgenes do not disrupt wild-type axon guidance (*Figure 5—figure supplement 1D*, *Tables 1 and 2*). With these transgenes, we aimed to determine whether the distinct axon behaviors at the vSLNC intermediate target and at VNC are mediated by UNC-6's chemotactic or haptotactic activity.

These transgenes are unable to restore ventral directionality of axon emergence (*Figure 5—figure supplement 1B, C*). Interestingly, stabilization at vSLNC is restored when non-diffusible UNC-6 is present at vSLNC (*Pmec-7::unc-6*) but not when expressed only at the VNC (*Punc-4::unc-6*), consistent with stabilization being contact-mediated (*Figure 5B, C*, *Videos 6–8*). Given the rapid and spatially specific accumulation of growth cone UNC-40 observed in the movies of PDE axon outgrowth, we next tested whether membrane-tethered UNC-6 could rescue UNC-40::GFP polarization in HSN. Indeed, non-diffusible UNC-6 along the vSLNC restored UNC-40::GFP polarity in HSN (*Figure 5D, E*). In contrast, non-diffusible UNC-6 expressed along the VNC could not restore ventral UNC-40 polarization (*Figure 5D, E*, *Figure 5—figure supplement 1E, F*), presumably due to the inability of HSN to contact this UNC-6 population. These data indicate that axonal stabilization at the vSLNC intermediate target requires local UNC-6, and membrane-tethered UNC-6 is sufficient for this function. Therefore, haptotaxis is sufficient to account for adhesion and UNC-40 polarization.

Next, we analyzed the directional filopodia growth in animals expressing membrane-tethered UNC-6. Interestingly, none of the transgenes were sufficient to restore biased ventral branching

(*Figure 5F*). This is consistent with a model where the biased direction of axon growth after adhesion at the vSLNC intermediate target requires diffusible UNC-6. Lastly, the frequency of axon stabilization at the VNC is partially rescued only in animals with membrane-tethered UNC-6 at both the vSLNC and VNC (*Figure 5G*). This result suggests that VNC stabilization is contact-mediated, but its efficiency requires UNC-6 at the intermediate target along earlier portions of the axon's path of growth. Taken together, these data suggest that contact-mediated haptotaxis drives axon stabilization at the vSLNC intermediate target and VNC, and that polarized UNC-40 toward the ventral cell body and growth cone is required for adhesion.

Given the ability of non-diffusible UNC-6 constructs to restore contact-mediated axon guidance behaviors, we considered their impact on the ultimate morphology of HSN and PDE neurons. By assaying adult neuronal morphology, we found that only the double transgene of membrane-tethered UNC-6 was sufficient to partially restore adult axonal morphology with 68.6% ± 5.0% of PDE axons and 47.4% ± 4.6% of HSN axons reaching the VNC by adulthood (*Tables 1 and 2*). This is in comparison to 100% of PDE axons and 99.1% ± 0.9% of HSN axons doing so in wild-type animals and 32.1% ± 5.1% of PDE axons and 33.1% ± 4.3% of HSN axons in *unc-6* mutants. These data indicate that membrane-tethered UNC-6 can partially rescue axon guidance when present at both vSLNC and VNC. This implies that diffusible UNC-6, possibly in the form of a gradient, is also required for axon guidance of PDE and HSN.

## UNC-6 is present in a shallow gradient

To further understand UNC-6's distribution and activity, we used an endogenous C-terminus mNeonGreen (mNG) tag of UNC-6 (UNC-6::mNG) (*Naegeli et al., 2017*) to visualize its localization in vivo. Our genetic evidence suggests that UNC-6 functions not only locally at vSLNC and VNC but also likely over the entire distance between VNC and vSLNC. We closely examined the localization of UNC-6::mNG in this region. We first imaged the emerging PDE growth cone interacting with UNC-6::mNG. In these images, bright mNG puncta surround the growth cone and its ventral path (*Figure 6A*). Notably, however, we are unable to detect an obvious gradient of UNC-6::mNG between the VNC and dorsal region of the animal.

We next imaged the zone between VNC and vSLNC where numerous faint UNC-6 puncta are present. Interestingly, bright UNC-6 puncta are also localized along the axonal filopodia of HSN. A similar localization along the axonal filopodia is also observed for PDE and AVM (*Figure 6B*). Such signal is completely absent in strains without the endogenous UNC-6::mNG tag, validating that the signal represents endogenous UNC-6 (*Figure 6—figure supplement 1*). We measured the intensity of UNC-6 through this region along the

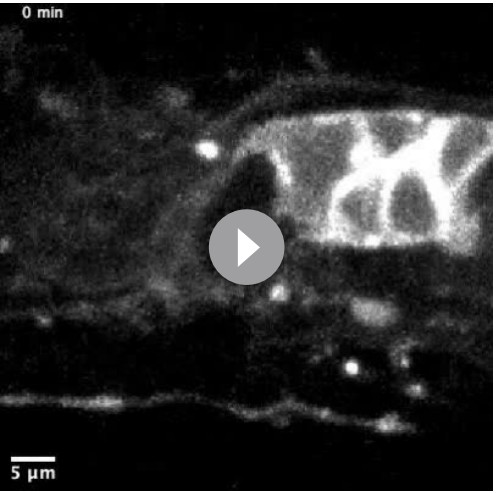

**Video 5.** PDE axon guidance in *madd-2(ok2226)* mutants.

https://elifesciences.org/articles/100424/figures#video5

**Video 6.** PDE axon guidance in *unc-6(ev400); Pmec-7::unc-6* animals.

https://elifesciences.org/articles/100424/figures#video6

ventrodorsal axis. The average intensity over this region is the highest at the VNC. Moving dorsally, the average intensity steadily decreases but increases again near the vSLNC (*Figure 6C*). These data are consistent with a model where UNC-6 is enriched in both the vSLNC and VNC regions separated by two shallow yet opposing gradients. Axon growth past the vSLNC intermediate target is likely due to permissive growth in response to high local concentrations of UNC-6, as recently proposed by *Hooper and Lundquist, 2024*.

Given the striking enrichment of UNC-6 along axonal filopodia, we hypothesized that UNC-40 is required for UNC-6 clustering along the filopodia. Animals carrying *unc-40* null mutations display a dorsoventral gradient similar to wild-type animals (*Figure 6D, E*). However, axonal filopodia that extend toward the VNC are not associated with UNC-6 puncta (*Figure 6D*). Taken together, these data suggest that UNC-6 is present in a gradient and decorates the periphery of the growth cone during growth.

## UNC-6 dynamically assemble around the growth cone during extension

The filopodia between vSLNC and VNC during axon outgrowth are dynamic, with constant growth and retraction. This implies that the UNC-6 puncta associated with the filopodia membrane are also likely not static. To directly test this, we simultaneously imaged five HSN growth cones and UNC-6::mNG puncta during its outgrowth in the region between the VSLNC and VNC. For this analysis, we only imaged growth cones that had extended into the space between the vSLNC and the VNC. Indeed, we observed dynamic assembly and disassembly of mNG puncta near the growth cone and filopodia on ~1 min time scales (*Figure 7A*, *Video 9*). In movies of *unc-40* null animals, these foci assemble but quickly disperse and do not associate with the growth cone (*Figure 7A*, *Video 10*). To quantify this, we calculated the intensity of each UNC-6 cluster adjacent to the cell membrane. These measurements reveal decreased UNC-6 labeling around 5 HSN growth cones in *unc-40* mutants compared to wild-type animals (*Figure 7B*). Axonal growth was rarely observed in *unc-40* mutants within the 120 s imaging window (*Figure 7A*), consistent with the reduced growth speed (*Figure 2I*). Together, these data suggest that there is a diffusive population of UNC-6 in the axonal path and that the extending growth cone remodels the UNC-6 landscape by assembling transient, highly concentrated foci around the cell to stimulate growth via UNC-40.

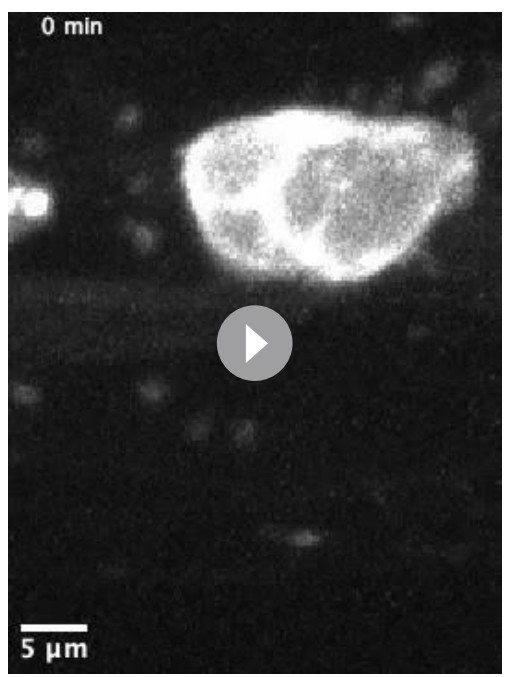

**Video 7.** PDE axon guidance in *unc-6(ev400); Punc-4::unc-6* animals.

https://elifesciences.org/articles/100424/figures#video7

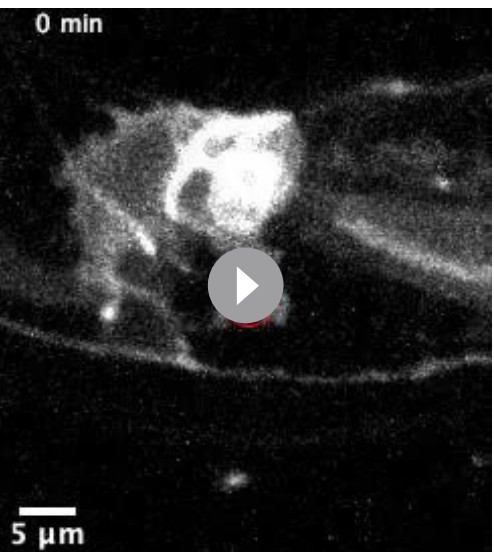

**Video 8.** PDE axon guidance in *unc-6(ev400); Pmec-7::unc-6; Punc-4::unc-6* animals.

https://elifesciences.org/articles/100424/figures#video8

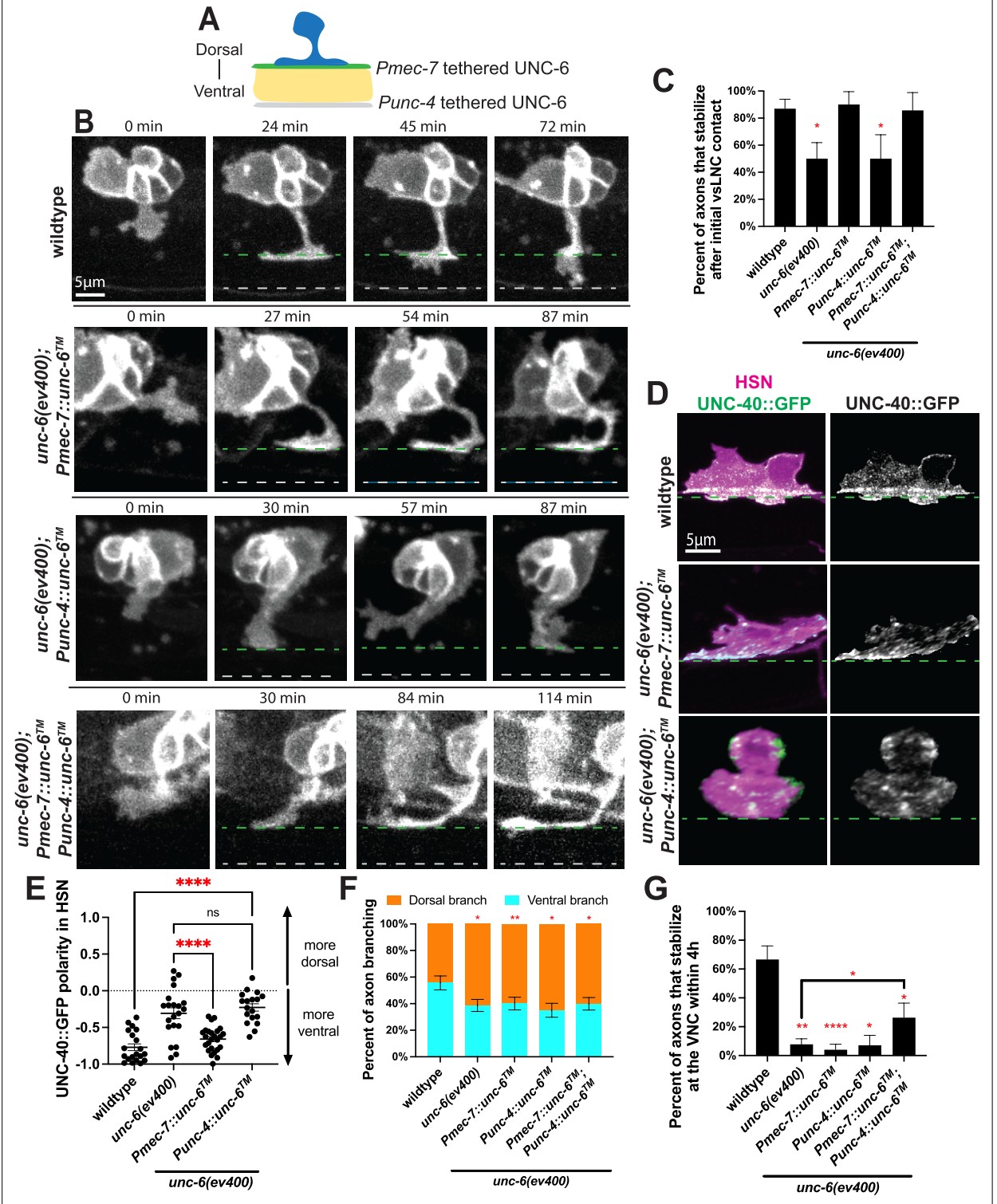

**Figure 5.** Diffusible UNC-6 is dispensable for axon stabilization. (**A**) Cartoon schematic of the non-diffusible rescue paradigms at the ventral sublateral nerve cord (vSLNC) (green, using *Pmec-7*) and at the ventral nerve cord (VNC) (magenta, using *Punc-4*). (**B**) Confocal time-lapse images of PDE axons guidance in wild-type, *unc-6(ev400); Pmec-7::unc-6*, *unc-6(ev400); Punc-4::unc-6* and *unc-6(ev400); Pmec-7::unc-6; Punc-4::unc-6* animals. Images were taken every 3 min for 4 hr in the L2 stage. Green lines denote the vSLNC and gray lines denote the VNC. Solid lines denote the location of membrane-tethered UNC-6. (**C**) Percentage of axons that stabilize in the sublateral region upon initial contact in wild-type (*n* = 23), *unc-6(ev400)* (*n* = 18), *unc-6(ev400); Pmec-7::unc-6* (*n* = 10), *unc-6(ev400); Punc-4::unc-6* (*n* = 8), and *unc-6(ev400); Pmec-7::unc-6; Punc-4::unc-6* (*n* = 7) animals. SE is shown. (**D**) Airyscan images of HSN axon emergence and endogenous UNC-40::GFP localization in wild-type, *unc-6(ev400), unc-6(ev400); Pmec-7::unc-6,* and

*Figure 5 continued on next page*

*Figure 5 continued*

*unc-6(ev400); Punc-4::unc-6* animals. (**E**) UNC-40::GFP polarity in HSN in wild-type (*n* = 20), *unc-6(ev400)* (*n* = 21), *unc-6(ev400); Pmec-7::unc-6* (*n* = 25), and *unc-6(ev400); Punc-4::unc-6* (*n* = 18) animals. Positive values denote dorsal polarization, and negative values denote ventral polarization. SEM is shown. (**F**) Proportion of branches formed dorsally (orange) and ventrally (cyan) from the sublateral region in wild-type (*n* = 90 branches, 24 animals), *unc-6(ev400)* (*n* = 114 branches, 25 animals), *unc-6(ev400); Pmec-7::unc-6* (*n* = 105 branches, 14 animals), *unc-6(ev400); Punc-4::unc-6* (*n* = 83 branches, 10 animals), and *unc-6(ev400); Pmec-7::unc-6; Punc-4::unc-6* (*n* = 108 branches, 15 animals) animals. SE is shown. (**G**) Percent of axons that stabilize that the VNC within the 4 hr imaging window in wild-type (*n* = 24), *unc-6(ev400)* (*n* = 25), *unc-6(ev400); Pmec-7::unc-6* (*n* = 14), *unc-6(ev400); Punc-4::unc-6* (*n* = 10), and *unc-6(ev400); Pmec-7::unc-6; Punc-4::unc-6* (*n* = 15) animals. SE is shown. Scale bars denote 5 µm in (**B, D**). Fisher's exact test with Bonferonni's correction is used in (**C, F, G**). Ordinary one-way ANOVA with multiple comparisons is used in (**E**). * denotes $p < 0.05$, ** denotes $p < 0.01$, **** denotes $p < 0.0001$. All comparisons to wild-type unless otherwise noted.

The online version of this article includes the following source data and figure supplement(s) for figure 5:

**Source data 1.** Quantification for *Figure 5*, *Figure 5—figure supplement 1*.

**Figure supplement 1.** Membrane-tethered UNC-6 is not sufficient to explain axon emergence.

## Discussion

Here, we use time-lapse imaging of in vivo growth cone behaviors to demonstrate that UNC-6/Netrin and UNC-40/DCC mediate both axon adhesion and directed growth. UNC-6 and UNC-40/DCC are required for stepwise axon adhesion and growth. Axon stabilization requires contact with UNC-6 which cooperates with the E3 ubiquitin ligase MADD-2 to polarize UNC-40, providing adhesive traction upon reaching an intermediate target. Biased growth requires UNC-6 and UNC-40 but cannot be explained by contact-dependent UNC-6 signaling. Indeed, we the highest concentrations of UNC-6 near the VNC which gradually decrease in the few microns dorsal to the VNC. Growth through this region requires capturing of UNC-6 clusters by UNC-40. These insights underscore the dynamic process of Netrin guidance as growth cones sequentially utilize haptotaxis and chemotaxis (*Figure 7C*).

### Complementary roles for haptotaxis and chemotaxis in UNC-6/Netrin guidance

Netrin has conserved functions in nervous system development across phylogeny. Initially, Netrin was proposed to function as a chemotactic cue that stimulates directional axon growth (*Kennedy et al., 1994*; *Serafini et al., 1994*). Indeed, secreted Netrin from the vertebrate floor plate functions to attract axons toward the ventral midline, avoiding the developing motor column (*Wu et al., 2019*). However, additional investigations in the vertebrate spinal cord as well as in the *Drosophila* visual system and midline have identified haptotactic roles for Netrin in axon guidance (*Akin and Zipursky, 2016*; *Brankatschk and Dickson, 2006*; *Varadarajan et al., 2017*). Our data clearly demonstrate roles for both chemotaxis and haptotaxis in Netrin-mediated attraction. Specifically, haptotaxis (mediated by membrane-tethered UNC-6) is sufficient to explain growth cone stabilization by providing adhesive force to maintain axonal contact with an intermediate target. Conversely, chemotaxis likely provides the basis of directional growth toward the VNC.

Importantly, the endogenous distribution of UNC-6/Netrin supports dual migration mechanisms of haptotaxis and chemotaxis. In *C. elegans*, the ventral sublateral intermediate target is separated from the VNC by a gradient of UNC-6 with the highest concentration along the VNC. Yet, we also observe that this gradient is modified by an accumulation of UNC-6 in the sublateral region, consistent with the location of the axon's intermediate target. However, this poses an additional conundrum where axons must extend past the vSLNC into a region with lower relative UNC-6 concentrations. One possibility is that UNC-6 acts as a permissive cue to stimulate growth past the vSLNC. Such permissive functions for UNC-6 have also been reported in repulsive axon guidance in *C. elegans* (*Hooper and Lundquist, 2024*).

The precise mechanisms which produce elevated levels of UNC-6 in the sublateral region remains unclear. As a laminin-related protein, Netrin is likely interacting with extracellular matrix (ECM) proteins, and the enrichment of an ECM-binding partner in the sublateral could restrain UNC-6 in such a way as to promote haptotaxis. One candidate class of molecules is heparin sulfate proteoglycans which have high affinity to UNC-6 (*Serafini et al., 1994*). In fact, structural analysis of the Netrin/DCC interaction has identified a putative binding site dependent on extracellular sulfate (*Finci et al., 2014*).

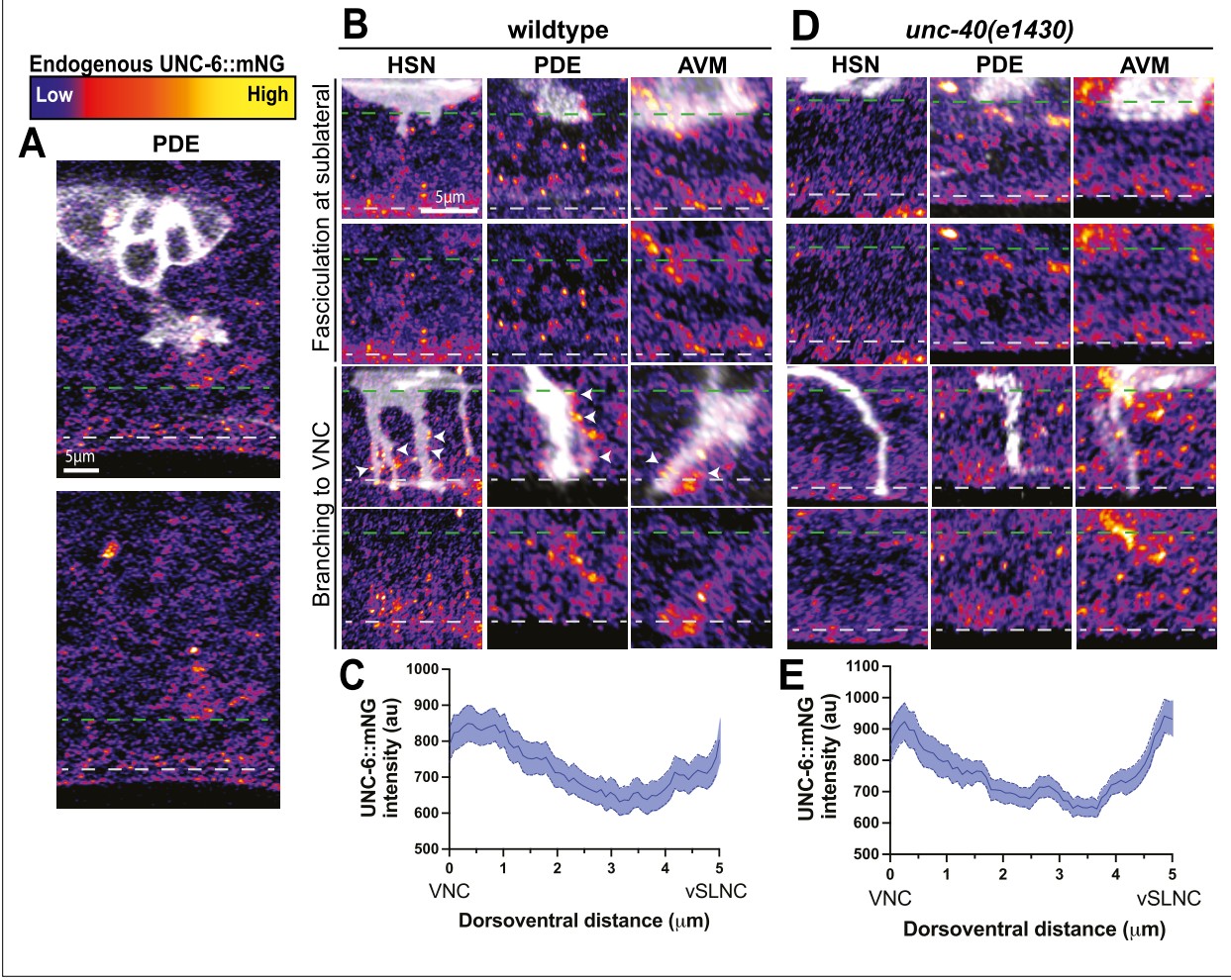

**Figure 6.** UNC-6 is present in a gradient. (**A**) Airyscan images of axon emergence in PDE and endogenous UNC-6::mNG distribution in wild-type animals. (**B**) Airyscan images of axon extension in HSN, PDE, and AVM and endogenous UNC-6::mNG distribution in wild-type animals. White arrowheads denote UNC-6 clusters around axonal filopodia. Dashed green line denotes ventral sublateral nerve cord (vSLNC) and dashed gray line denotes ventral nerve cord (VNC). (**C**) Average intensity of UNC-6::mNG between the VNC (left) and vSLNC (right) in wild-type animals (n = 49 animals). (**C**) Airyscan images of axon extension in HSN, PDE, and AVM and endogenous UNC-6::mNG distribution in *unc-40(e1430)* animals. Dashed green line denotes vSLNC and dashed gray line denotes VNC. (**D**) Average intensity of UNC-6::mNG between the VNC (left) and vSLNC (right) in *unc-40(e1430)* animals (n = 37 animals). SEM is shown. Scale bars denote 5 μm in (**A, B**).

The online version of this article includes the following source data and figure supplement(s) for figure 6:

**Source data 1.** Quantification for *Figure 6*.

**Figure supplement 1.** Levels of autofluorescence in the region between the ventral sublateral nerve cord (vSLNC) and ventral nerve cord (VNC) are low.

A recent study in *C. elegans* supports these conclusions using a transmembrane-tethered allele of *unc-6* (*Hooper and Lundquist, 2024*). This allele displays a partial loss-of-function phenotype that can be restored through UNC-6 produced by an ectopic source. These data suggest that the diffusibility of UNC-6 is more critical than the location of its expression in *C. elegans*. Therefore, the gradient that we observe is likely established using local interactions between UNC-6 and the surrounding ECM. Additional studies combining cell biological and structural information may provide further insight on UNC-6 interactions with the ECM and the dichotomous signaling mechanisms that promote hapto-tactic and chemotactic responses.

## Regulation and localization of UNC-40/DCC is crucial to haptotaxis

Haptotactic growth cone adhesion serves a central function in growth cone migrations (*Comer et al., 2019*). Investigations of the molecular signaling mechanisms that promote Netrin-driven haptotaxis

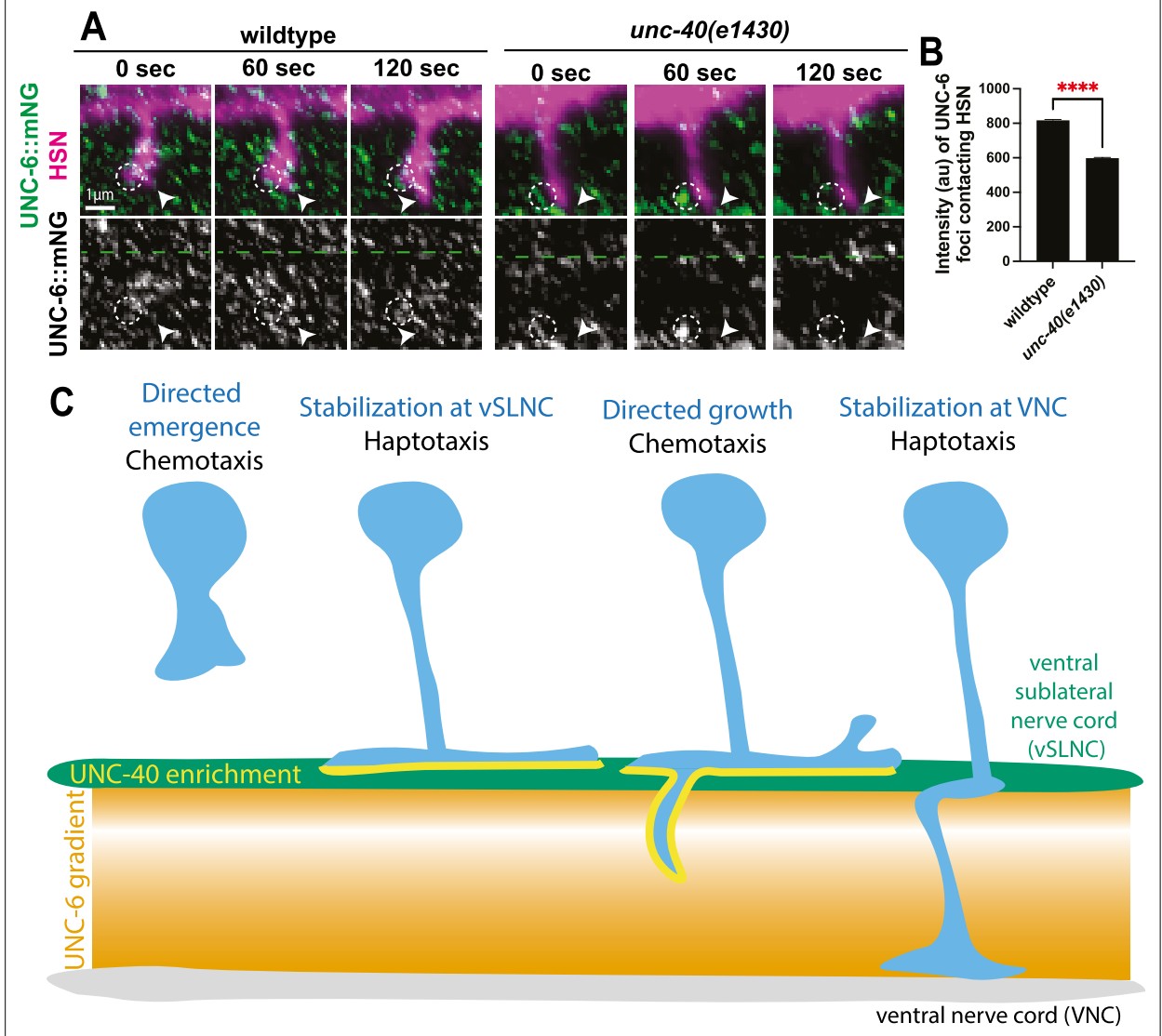

**Figure 7.** UNC-6 clusters are dynamically assembled at the growth cone tip during axon extension. (**A**) Airyscan time-lapse images of HSN axon extension and endogenous UNC-6::mNG in wild-type and *unc-40(e1430)* animals. Images were taken every 30 s for 30 min. White arrowhead denotes the growth cone tip. Dashed white circle denotes UNC-6 clusters on the lateral edges of the growth cone. (**B**) Intensity of UNC-6 foci adjacent to the growth cone membrane in wild-type (*n* = 861 foci, 5 animals) and *unc-40(e1430)* (*n* = 1306 foci, 5 animals) animals. (**C**) Model of ventral axon migration via alternating haptotaxis and chemotaxis. Scale bar denotes 1 μm in (**A**). Unpaired Student's *t*-test is used in (**B**). **** denotes p < 0.0001.

The online version of this article includes the following source data for figure 7:

**Source data 1.** Quantification for *Figure 7*.

have implicated an 'adhesion-clutch model' where DCC generates force through connections to the actin cytoskeleton (*Qiu et al., 2024*). Our data identify another molecular regulatory feature of haptotaxis: guidance receptor localization. In ventrally migrating axons, endogenous levels of UNC-40 receptor are polarized toward the ventral side of the cell upon contact with the sublateral intermediate target. The E3 ubiquitin ligase MADD-2/Trim9/Trim67 cooperates with UNC-6 in this polarization. These results are consistent with cell invasion mechanisms in the non-neuronal anchor cell in *C. elegans* (*Wang et al., 2014*). In our neuronal system, genetic paradigms that disrupt UNC-40 polarization also exhibit decreased adherence to the sublateral region (*Figures 4 and 5*). Therefore, we posit that polarization of UNC-40 likely amplifies the cell's capacity to adhere to its intermediate target.

Given our data suggesting that MADD-2 ubiquitinates UNC-40, we propose that ubiquitination of UNC-40 by MADD-2 triggers UNC-40 endocytosis. Similar data in vertebrate systems identify that

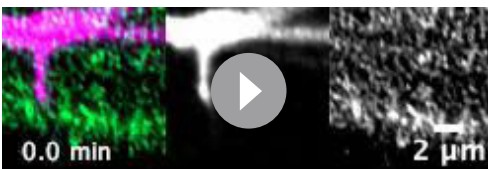

**Video 9.** HSN growth cone interactions with UNC-6::mNG in wild-type animals.
https://elifesciences.org/articles/100424/figures#video9

**Video 10.** HSN growth cone interactions with UNC-6::mNG in *unc-40(e1430)* animals.
https://elifesciences.org/articles/100424/figures#video10

Netrin antagonizes Trim9-induced ubiquitination of DCC (*Plooster et al., 2017*). Together, these data suggest that in these neurons MADD-2 likely functions in areas where UNC-6 concentration is low (i.e. the dorsal region of the cell/growth cone). In this model, UNC-40 would be preferentially endocytosed in dorsal regions of the cell, thereby contributing to the overall polarity of UNC-40. In contrast, UNC-6 likely stabilizes UNC-40 expression on the cell membrane and prevents ubiquitination at the ventral side of the cell, further establishing UNC-40 polarity. Consistently, experiments in vertebrate systems have long suggested that exposure to Netrin causes DCC to be added to the cell membrane through exocytosis (*Matsumoto and Nagashima, 2010*; *Plooster et al., 2017*).

Notably, *Drosophila* photoreceptors depend on Trim9 for axon stabilization but not elongation in Netrin guidance, consistent with our movies of *madd-2* mutant axon guidance (*Akin and Zipursky, 2016*). Additionally, studies of vertebrate neurons demonstrate that Trim9 ubiquitinates DCC when the concentration of Netrin is low to regulate DCC exocytic balance (*Plooster et al., 2017*). Given the requirement of four juxtamembrane lysine residues in UNC-40 polarization, our data suggest that this regulatory function of MADD-2/Trim9 is conserved and crucial for UNC-40 cell polarity and growth cone adhesion. While the data here do not directly examine cytoskeletal regulation in relation to haptotaxis, Trim9 and Trim 67 are known to regulate members of the Ena/VASP family through ubiquitination to control filopodia density in response to Netrin-1 (*Boyer et al., 2020*; *McCormick et al., 2024*; *Menon et al., 2015*). Additionally, the lone Ena/VASP family member in *C. elegans* (UNC-34) is required for ventral axon guidance downstream of UNC-40 (*Gitai et al., 2003*; *Norris et al., 2009*). Systematic investigations of the interplay between ubiquitination, receptor localization, and cytoskeletal dynamics are required to fully elaborate haptotactic signaling mechanisms in growth cones.

Collectively, our data provide spatiotemporal access to Netrin-mediated attraction. During growth, the axon alternates between haptotactic and chemotactic migration, and we delineate the precise axonal behaviors controlled by each migration mechanism. Overall, this study predicates future investigations of the precise molecular cascades that govern discrete chemotactic and haptotactic responses to Netrin within the same growth cone.

## Materials and methods

### Key resources table

| Reagent type (species) or resource | Designation | Source or reference | Identifiers | Additional information |
|---|---|---|---|---|
| strain, strain background (*C. elegans*) | See *Supplementary file 1* | See *Supplementary file 1* | See *Supplementary file 1* | |
| cell line (*E. coli*) | OP50 | CGC | CGC: OP50; RRID:WB-STRAIN:WBStrain00041969 | |
| recombinant DNA reagent | pELN3 (plasmid) | this paper | N/A | Pmec-7::unc-6 cDNA::nlg-1 transmembrane domain::SNAP |
| recombinant DNA reagent | pELN8 (plasmid) | this paper | N/A | Punc-4::unc-6 cDNA::nlg-1 transmembrane domain::SNAP |
| sequence-based reagent | UNC-40::GFP::AID sgRNA | IDT | this paper | Sequence: ATTTGCATTTTGAGACGAGTA |

*Continued on next page*

*Continued*

| Reagent type (species) or resource | Designation | Source or reference | Identifiers | Additional information |
|---|---|---|---|---|
| sequence-based reagent | GLO-1 sgRNA | IDT | this paper | Sequence: GAAAACATTAGTAATTTAAA |
| sequence-based reagent | UNC-40(4KR) sgRNA #1 | IDT | this paper | Sequence: ATTTGCATTTTGAGACGAGTA |
| sequence-based reagent | UNC-40(4KR) sgRNA #2 | IDT | this paper | Sequence: TCTGGAAAAAAGACGAGTGC |
| sequence-based reagent | Alt-R CRISPR-Cas9 tracrRNA | IDT | IDT: 1072532 | |
| peptide, recombinant protein | Alt-R S.p.Cas9 Nuclease V3 | IDT | IDT: 1081059 | |
| commercial assay or kit | PrimeSTAR GXL DNA Polymerase | Takara | Takara: 050B | |
| commercial assay or kit | Gibson Assembly Master Mix | NEB | NEB: E2611S | |
| chemical compound, drug | Levamisole | Sigma-Aldrich | Sigma-Aldrich: 1359302; CID: 26879 | |
| software, algorithm | Fiji | NIH | RRID:SCR_003070 | https://imagej.nih.gov/ij/ |
| software, algorithm | mTrackJ | Image Science | RRID:SCR_001337 | https://imagescience.org/meijering/software/mtrackj/ |
| software, algorithm | Prism 10 | GraphPad | RRID:SCR_002798 | https://www.graphpad.com/scientific-software/prism/ |
| software, algorithm | Python | Python Software Foundation | RRID:SCR_008394 | https://www.python.org/ |
| software, algorithm | Plotly v5.22.0 | Plotly | RRID:SCR_013991 | https://plotly.com/ |

## Resource availability

### Lead contact

Further information and requests for resources and reagents should be directed to and will be fulfilled by the lead contact, Kang Shen (kangshen@stanford.edu).

### Materials availability

All materials used and generated in this study, including plasmids and worm strains, are available for sharing by contacting Kang Shen (kangshen@stanford.edu).

## Experimental model and subject details

### *C. elegans* methods

All *C. elegans* animals used in this study were handled at 20°C on NGM plates with OP50 *E. coli* as food, according to standard protocols (*Brenner, 1974*). N2 Bristol was used as a wild-type reference strain. Transgenic strains were generated via microinjection of 1–5 ng/μl of plasmid DNA and of 1 ng/μl of selection markers plasmids (*Pmyo-2::mCherry*). Pharyngeal mCherry signal was used to identify transgenic worms. All analyses were performed on *C. elegans* hermaphrodites. Imaging of axon outgrowth used animals aged to the L2 stage. Imaging of mature neuronal morphologies used animals aged to the L4 and young adult stages. Detailed description of the strains used in this study are available in *Supplementary file 1*.

## Method details

### Confocal imaging

All images were taken at room temperature using live *C. elegans*. L2 animals were anesthetized using 6 mM levamisole in M9 buffer and mounted on 4% agarose pads. For still images, worms were imaged on an inverted Zeiss Axio Observer Z1 microscope with a Yokogawa CSU-X1 spinning-disk

unit, a Hamamatsu EM-CCD digital camera controlled by Metamorph (version 7.8.12.0). Images were acquired using a Plan-Apochromat 100× 1.4 NA objective. For dynamic imaging of axon outgrowth, worms were imaged on an inverted Zeiss Axio Observer Z1 microscope with a Yokogawa CSU-W1 spinning-disk unit, a Prime 95B Scientific CMOS camera controlled by 3i Slidebook (v6). Images were acquired using a C-Apochromat 63× 1.2 NA objective. Images were acquired every 3 min for 4 hr. For long-term dynamic imaging, the coverslip was sealed on all sides using valap to prevent sample dehydration.

### Airyscan imaging

All images were taken at room temperature using live *C. elegans*. L2 animals were anesthetized using 6 mM levamisole in M9 buffer and mounted on 4% agarose pads. Worms were imaged on a Zeiss LSM 980 Airyscan 2 system controlled by Zen (version 3.5 blue edition). Images were acquired using a Plan-Apochromat 63× 1.4 NA objective. For dynamic imaging of UNC-6::mNG, images were acquired every 30 s for 30 min.

### Constructs and cloning

All constructs used in this study are listed in the Key Resources Table. Plasmid constructs generated for this study were created using isothermal assembly with overlapping oligonucleotides (*Gibson et al., 2009*). Expression constructs were generated in a pSM delta vector. Non-diffusible UNC-6 constructs were made by fusing the *nlg-1* transmembrane domain sequence to the C-terminus of the *unc-6* coding sequence. A SNAP tag was fused to the intracellular side of the *nlg-1* transmembrane domain.

### Germline editing using CRISPR/Cas9

Endogenous fluorophore insertions and base pair changes were generated by gonadal microinjection of CRISPR/Cas9 protein complexes. Genome editing was performed using standard protocols (*Dokshin et al., 2018*). Cas9 protein, tracRNA, and crRNAs were injected at a concentration of 1.525 µM. crRNAs used in this study are available in the Key Resources Table. DNA repair templates were created by PCR or ordered as ultramers and injected at 2 µM. pRF4 (*rol-6(su1006)*) was used as a co-injection marker and selected against in the $F_2$ generation. $F_2$ animals were screened for the desired edits using PCR and confirmed by sequencing. Newly generated strains with the proper edits were outcrossed five times before use in experiments.

### Electron microscopy

N2 Bristol worms staged to L4 were prepared for conventional EM by high-pressure freezing/freeze-substitution. Worms were suspended in M9 containing 20% BSA and *E. coli* and frozen in 100 µm well carriers (Type A) across from a hexadecane-coated flat carrier (Type B) with a BalTec HPM 01 high-pressure freezer. A Leica AFS2 unit was used for freeze-substitution in 1% $OsO_4$, 0.1% uranyl acetate, 1% methanol in acetone, and 3% water. After substitution, the samples were rinsed in acetone, infiltrated and polymerized in Eponate 12 resin. A Leica UCT ultramicrotome with a Diatome diamond knife was used to cut serial 50 nm sections. The sections were picked up on Pioloform film-coated slot grids and stained with uranyl acetate and Sato's lead. An FEI Tecnai T12 TEM at 120 kV with a Gatan 4k × 4k U895 camera was used to image the sections. Serial sections were aligned using TrakEM2 in Fiji.

### UNC-40::GFP visualization

Images of UNC-40::GFP in single cells were generated by masking using Fiji. In short, the image channel of the cell membrane marker was used to generate a binary mask. This binary mask was applied to the UNC-40::GFP channel, removing fluorescence not associated with the cell membrane mask and retaining the original intensity values of UNC-40::GFP. This protocol was used to generate all of the UNC-40::GFP images in this manuscript.

### SNAP labeling

SNAP labeling was performed consistent with previous protocols in *C. elegans* (*Kurshan et al., 2018*). Worms were collected from the plate by washing with M9 buffer. The collected worms were then spun down and the supernatant was removed. This washing process was repeated five times. Next, the worms were washed in a total volume of 25 µl with 2 mM SNAP-Cell Oregon Green dye (New England Biosciences). The sample was rotated in this solution for 3 hr. Following this staining, the sample was washed five times in 1 ml M9 buffer. The worms were then transferred to a food-stocked agar plate overnight. Images were taken 16 hr later.

## Quantification and statistical analysis

### UNC-40::GFP polarity measurements

Fiji was used to measure the integrated fluorescent density along the cell dorsal and ventral periphery in single z-planes as visualized in *Figure 2F*. The ventral integrated density was subtracted from the dorsal integrated density, and this value was divided by the total integrated density in the ventral and dorsal regions. This calculated value is reported as the polarity of the fluorescent marker. All measurements were taken in raw images prior to processing of UNC-40::GFP for visualization as described above.

### Analysis of dynamic imaging of axon growth

Fiji was used to analyze movies of axon outgrowth. Growth cone and UNC-40::GFP tracings were completed using the MTrackJ plugin on Fiji. Intensity values were normalized to the traced intensity in the growth cone upon contact with the sublateral region. All other analyses were reported by manual counting or length measurements. Dorsal and ventral branching were determined by protrusions longer than 0.6 µm. Growth speed was calculated as the total displacement of the branch divided by its duration in the movies.

### Analysis of dynamic imaging of UNC-6::mNG

Fiji was used to analyze movies of UNC-6::mNG dynamics. First, UNC-6::mNG foci overlapping with the cell membrane marker were isolated via the masking method described above for UNC-40::GFP, retaining the UNC-6::mNG intensity values. Next, z-projections were made of the movies showing UNC-6::mNG contacts with the cell membrane. Last, the Analyze Particles analysis on Fiji was run to generate the average intensity of each identified UNC-6::mNG cluster.

### Data visualization and statistical analysis

Plotly (via Python) was used to generate the roseplots. GraphPad Prism 10 was used for all other data visualization and statistical analyses. SEM and SE are shown as indicated in figure legends. Statistical analyses used are also available in figure legends. Data collection and analysis were not performed blind to experimental conditions.

## Acknowledgements

We thank the members of the Shen Lab for their feedback on the study and on the manuscript. We particularly thank Richard Fetter, Callista Yee, Kelsie Eichel, and Dane Kawano for their detailed input on the study and feedback on the manuscript. We also thank Richard Fetter for *Figure 1D*. This work was funded by the Howard Hughes Medical Institute (where K S is an investigator) and the NSF Graduate Research Fellowship Program (E L N).

## Additional information

### Funding

| Funder | Grant reference number | Author |
|---|---|---|
| Howard Hughes Medical Institute | | Kang Shen |
| National Institutes of Health | R01NS082208 | Kang Shen |
| National Science Foundation | Graduate Research Fellowship Program | Ev L Nichols |

The funders had no role in study design, data collection and interpretation, or the decision to submit the work for publication.

### Author contributions

Ev L Nichols, Conceptualization, Resources, Data curation, Software, Formal analysis, Validation, Visualization, Methodology, Writing – original draft, Writing – review and editing; Joo Lee, Methodology; Kang Shen, Conceptualization, Supervision, Funding acquisition, Project administration

### Author ORCIDs

Ev L Nichols ⓘ https://orcid.org/0000-0001-9835-9107
Joo Lee ⓘ https://orcid.org/0000-0001-5875-6036
Kang Shen ⓘ https://orcid.org/0000-0003-4059-8249

Reviewer #1 (Public review): https://doi.org/10.7554/eLife.100424.3.sa1
Reviewer #2 (Public review): https://doi.org/10.7554/eLife.100424.3.sa2
Reviewer #3 (Public review): https://doi.org/10.7554/eLife.100424.3.sa3
Author response https://doi.org/10.7554/eLife.100424.3.sa4

## Additional files

### Supplementary files

Supplementary file 1. Summary of *C. elegans* strains used in this study.

MDAR checklist

### Data availability

All data generated or analyzed during this study are included in the manuscript and supporting files, source data files have been provided.

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
