## [Editor Report · eLife Assessment]

These studies make a **fundamental** contribution to our understanding of axon-guidance mechanisms, focusing on the role of UNC-6/Netrin in the long-range growth and targeting of axons. Using state-of-the-art genetics and in vivo imaging, the authors provide **compelling** support for the finding that UNC-6/Netrin can act via both chemotaxis and haptotaxis. This work will be of interest to a wide variety of cell and developmental biologists and neuroscientists.

---

## [Referee Report · Reviewer #1 (Public review)]

Summary:

This paper investigates the mechanism of axon growth directed by the conserved guidance cue UNC-6/Netrin. Experiments were designed to distinguish between alternative models in which UNC-6/Netrin functions as either a short range (haptotactic) cue or a diffusible (chemotactic) signal that steers axons to their final destinations. In each case, axonal growth cones execute ventrally directed outgrowth toward a proximal source of UNC-6/Netrin. This work concludes that UNC-6/Netrin functions as both a haptotactic and chemotactic cue to polarize the UNC-40/DCC receptor on the growth cone membrane facing the direction of growth. Ventrally directed axons initially contact a minor longitudinal nerve tract (vSLNC) at which UNC-6/Netrin appears to be concentrated before proceeding in the direction of the ventral nerve cord (VNC) from which UNC-6/Netrin is secreted. Time lapse imaging revealed that growth cones appear to pause at the vSLNC before actively extending ventrally directed filopodia that eventually contact the VNC. Growth cone contacts with the vSLNC were unstable in unc-6 mutants but were restored by expression of a membrane tethered UNC-6 in vSLNC neurons. In addition, expression of membrane tethered UNC-6/Netrin in the VNC was not sufficient to rescue initial ventral outgrowth in an unc-6 mutant. Finally, dual expression of membrane tethered UNC-6/Netrin in both vSLNC and VNC partially rescued the unc-6 mutant axon guidance defect, thus suggesting that diffusible UNC-6 is also required. This work is important because it potentially resolves the controversial question of how UNC-6/Netrin directs axon guidance by proposing a model in which both of the competing mechanisms, e.g., haptotaxis vs chemotaxis, are successively employed. The impact of this work is bolstered by its use of powerful imaging and genetic methods to test models of UNC-6/Netrin function in vivo thereby obviating potential artifacts arising from in vitro analysis.

Strengths:

A strength of this approach is the adoption of the model organism *C. elegans* to exploit its ready accessibility to live cell imaging and powerful methods for genetic analysis.

Weaknesses:

In the revised version of this manuscript, the authors have redressed the weaknesses highlighted in my review of the original paper.

---

## [Referee Report · Reviewer #2 (Public review)]

Nichols et al studied the role of axon guidance molecules and their receptors and how these work as long-range and/or local cues, using in-vivo time-lapse imaging in *C. elegans*. They found that the Netrin axon guidance system, work in different modes when acting as a long-range (chemotaxis) cue vs local cue (haptotaxis). As an initial context, they take advantage of the postembryonic-born neuron, PDE, to understand how its axon grows and then is guided into its target. They found that this process occurs in various discrete steps, during which the growth cone migrates and pauses at specific structures, such as the vSLNC. The role of the UNC-6/Netrin and UNC-40/DCC axon guidance ligand-receptor pair was then looked at in terms of its requirement for (1) initial axon outgrowth direction, (2) stabilization at the intermediate target, (3) directional branching from the sublateral region or (4) ventral growth from intermediate target to the VNC. They found that each step is disrupted in the unc-6/Netrin and unc-40/DCC mutants and observed how the localization of these proteins changed during the process of axon guidance in wild type and mutant contexts. These observations were further supported by analysis of a mutant important for the regulation of Netrin signaling, the E3 ubiquitin ligase madd-2/Trim9/Trim67. Remarkably, the authors identified that this mutant affected axonal adhesion and stabilization, but not directional growth. Using membrane-tethered UNC-6 to specific localities, they then found this to be a consequence of the availability of UNC-6 at specific localities within the axon growth path. Altogether, this data and in-vivo analysis provide compelling evidence of the mechanistic foundation of Netrin-mediated axon guidance and how it works step by step.

The conclusions are well-supported, with both imaging and quantification of each step of axon guidance and localization of UNC-6 and UNC-40. Using a different type of neuron to validate their findings further supports their conclusions and strengthens their model. They also probe the role of the axon guidance ligand-receptor pair SLT-1/Slit and SAX-3/ROBO in this process and find it to work in parallel to UNC-6. This work sets up the stage for future analysis of other axon guidance molecules or regulators using time-lapse in-vivo imaging to better understand their role as long-range and/or local cues.

---

## [Referee Report · Reviewer #3 (Public review)]

Summary:

This manuscript from Nichols, Lee, and Shen tackles an important question of how unc6/netrin promotes axon guidance: i.e. haptotaxis vs chemotaxis. This has recently been a large topic of investigation and discussion in the axon guidance field. Using live cell imaging of unc6/netrin and unc40/DCC in several neurons that extend axons ventrally during development, as well as TM localized mutants of Unc6, they suggest that unc6 promotes first haptotaxis of the emerging growth cone followed by chemotaxis of the growth cone. This is timely, as a recent preprint from the Lundquist group, using a similar strategy to make only a TM anchored unc6 similarly found that this could rescue only the haptotaxis like growth of the PDE neuron, but not the second phase of growth. However, their conclusions were quite different based on the overexpression of unc6 everywhere rescuing the second phase, and thus they conclude that a gradient is not present.

Strengths:

As this has been quite a controversy in both the invertebrate and vertebrate fields, one strength of this paper is that they use a unc6-neon green to demonstrate unc6 localization, and show localization. Further, they provide localisation of the transmembrane tether version of netrin, showing its restriction to nerve cords.

---

## [Author Response]

The following is the authors’ response to the original reviews.

**Reviewer #1 (Public Review):**
Summary:This paper investigates the mechanism of axon growth directed by the conserved guidance cue UNC-6/Netrin. Experiments were designed to distinguish between alternative models in which UNC-6/Netrin functions as either a short-range (haptotactic) cue or a diffusible (chemotactic) signal that steers axons to their final destinations. In each case, axonal growth cones execute ventrally directed outgrowth toward a proximal source of UNC-6/Netrin. This work concludes that UNC-6/Netrin functions as both a haptotactic and chemotactic cue to polarize the UNC-40/DCC receptor on the growth cone membrane facing the direction of growth. Ventrally directed axons initially contact a minor longitudinal nerve tract (vSLNC) at which UNC-6/Netrin appears to be concentrated before proceeding in the direction of the ventral nerve cord (VNC) from which UNC-6/Netrin is secreted. Time-lapse imaging revealed that growth cones appear to pause at the vSLNC before actively extending ventrally directed filopodia that eventually contact the VNC. Growth cone contacts with the vSLNC were unstable in unc-6 mutants but were restored by the expression of a membrane-tethered UNC-6 in vSLNC neurons. In addition, the expression of membrane-tethered UNC-6/Netrin in the VNC was not sufficient to rescue initial ventral outgrowth in an unc-6 mutant. Finally, dual expression of membrane-tethered UNC-6/Netrin in both vSLNC and VNC partially rescued the unc-6 mutant axon guidance defect, thus suggesting that diffusible UNC-6 is also required. This work is important because it potentially resolves the controversial question of how UNC-6/Netrin directs axon guidance by proposing a model in which both of the competing mechanisms, e.g., haptotaxis vs chemotaxis, are successively employed. The impact of this work is bolstered by its use of powerful imaging and genetic methods to test models of UNC-6/Netrin function in vivo thereby obviating potential artifacts arising from in vitro analysis.Strengths:A strength of this approach is the adoption of the model organism *C. elegans* to exploit its ready accessibility to live cell imaging and powerful methods for genetic analysis.Weaknesses:A membrane-tethered version of UNC-6/Netrin was constructed to test its haptotactic role, but its neuron-specific expression and membrane localization are not directly determined although this should be technically feasible. Time-lapse imaging is a key strength of multiple experiments but only one movie is provided for readers to review.

Thank you for your comments. We have now used SNAP labeling to directly visualize the localization of membrane tethered UNC-6 and confirmed UNC-6 is only detectable on the sublateral and ventral nerve cords (Figure S3A). These data have been added to the manuscript on page 15, lines 342-347. We have also provided a representative movie for each imaged genotype (Videos S2-10).

**Reviewer #2 (Public Review):**
Nichols et al studied the role of axon guidance molecules and their receptors and how these work as long-range and/or local cues, using in-vivo time-lapse imaging in *C. elegans*. They found that the Netrin axon guidance system works in different modes when acting as a long-range (chemotaxis) cue vs local cue (haptotaxis). As an initial context, they take advantage of the postembryonic-born neuron, PDE, to understand how its axon grows and then is guided into its target. They found that this process occurs in various discrete steps, during which the growth cone migrates and pauses at specific structures, such as the vSLNC. The role of the UNC-6/Netrin and UNC-40/DCC axon guidance ligand-receptor pair was then looked at in terms of its requirement for(1) initial axon outgrowth direction(2) stabilization at the intermediate target(3) directional branching from the sublateral region or(4) ventral growth from the intermediate target to the VNC.They found that each step is disrupted in the unc-6/Netrin and unc-40/DCC mutants and observed how the localization of these proteins changed during the process of axon guidance in wild-type and mutant contexts. These observations were further supported by analysis of a mutant important for the regulation of Netrin signaling, the E3 ubiquitin ligase madd-2/Trim9/Trim67. Remarkably, the authors identified that this mutant affected axonal adhesion and stabilization, but not directional growth. Using membrane-tethered UNC-6 to specific localities, they then found this to be a consequence of the availability of UNC-6 at specific localities within the axon growth path. Altogether, this data and in-vivo analysis provide compelling evidence of the mechanistic foundation of Netrin-mediated axon guidance and how it works step by step.The conclusions are well-supported, with both imaging and quantification of each step of axon guidance and localization of UNC-6 and UNC-40. Using a different type of neuron to validate their findings further supports their conclusions and strengthens their model. It's not yet known whether this model holds true for other ligand-receptor pairs, but the current work sets the stage for future analysis of other axon guidance molecules using time-lapse in-vivo imaging. There are still two outstanding questions that are important to address to support the authors' model and conclusions.(1) The results of UNC-6-TM expression at different locations are clear and support the conclusions but need to consider that there's no diffusible UNC-6 available. What would happen if UNC-6 is tethered to the membrane in an otherwise completely 'normal' UNC-6 gradient. Does the axon guidance ensue normally or does it get stuck in the respective site of the membrane tethered-UNC-6 and doesn't continue to outgrow properly? This is an important control (expression of the UNC-6-TM at the vSLNC or VNC in the wild type background) that would help clarify this question and gain a better insight into the separability of both axon guidance steps and the ability to manipulate these.

Thank you for your comments. We expressed UNC-6 at vSLNC and VNC in wild-type animals and examined adult morphology of both HSN and PDE in the control conditions you suggested. These data are available in Tables 1 and 2 with no statistical differences compared to wildtype animals. Second, we also provide still images of developing PDE axons near the vSLNC (Figure S3D) to confirm that this axon guidance step is intact when UNC-6 is overexpressed in specific regions. Together, these data suggest that the TM rescue constructs do not interfere with endogenous axon guidance pathways. We have added these results to the manuscript on page 15, lines 347-349.

(2) Axon guidance systems do not work in a vacuum and are generally competing against each other. For example, the SLT-1/Slit and SAX-3/ROBO axon guidance ligand-receptor pair is also required for PDE, and other post-embryonic neurons, axon guidance. It would be interesting to test mutants for these genes with the membrane tethered-UNC-6 to determine if the different steps of axon guidance are disrupted and if so, to what degree these are disrupted.

Thank you for this suggestion. We have performed time-lapse imaging on *slt-1* mutants and *unc-6; slt-1* double mutants. These data are available in a new figure, Figure 3. Indeed, we found that *slt-1* mutants showed abnormal direction of axon emergence and stabilization at the VNC but normal stabilization at vsLNC and axonal branching (Fig.3). These data can be found in the manuscript from pages 11-12, lines 248-269.

**Reviewer #3 (Public Review):**
Summary:This manuscript from Nichols, Lee, and Shen tackles an important question of how unc6/netrin promotes axon guidance: i.e. haptotaxis vs chemotaxis. This has recently been a large topic of investigation and discussion in the axon guidance field. Using live cell imaging of unc6/netrin and unc40/DCC in several neurons that extend axons ventrally during development, as well as TM localized mutants of Unc6, they suggest that unc6 promotes first haptotaxis of the emerging growth cone followed by chemotaxis of the growth cone. This is timely, as a recent preprint from the Lundquist group, using a similar strategy to make only a TM anchored unc6 similarly found that this could rescue only the haptotaxis-like growth of the PDE neuron, but not the second phase of growth. However, their conclusions were quite different based on the overexpression of unc6 everywhere rescuing the second phase, and thus they conclude that a gradient is not present.Strengths:As this has been quite a controversy in both the invertebrate and vertebrate field, one strength of this paper is that they use an unc6-neon green to demonstrate unc6 localization, and show a gradient of localization.Weaknesses:This is important, although it could be strengthened by first showing a more zoomed-out image of unc6 in the animal, and second demonstrating the localization of the transmembrane anchored unc6 mutants, to help define what may be the "diffusible Unc6".

Thank you for your comments. We have performed both of these experiments. In Figure 6A, we provide a zoomed out image of PDE growth cone interacting with UNC-6::mNG prior to reaching the vSLNC. Notably, we do not observe an obvious gradient that extends into this more dorsal region of the animal. We have also shown the membrane localization of UNC-6 through SNAP labeling in Figure S3A. These data have been added to the manuscript on page 15, lines 342-347.

I suggest two additional experimental or analysis suggestions: First, the authors clarify the phenotype of ventral emergence of the growth cone. Though the manuscript images suggest that no matter the mutant there is ventral emergence of the growth cone, but then later defects, yet they claim ventral emergence defects with the UNC6 tethered mutants, but there is no comparison of rose plots. This is confusing and needs to be addressed.

Thank you for your comment. We have now included images (i.e. *slt-1(eh15)* and *unc-6(ev400); slt-1(eh15)* genotypes in Figure 3) and movies showing misoriented axon emergence. We have also provided an additional quantification that allows for statistical comparison of emergence angle across genotypes. This quantification takes the sine function of the angle to quantify the relative emergence trajectory across the dorsal-ventral axis. A value of 1 indicates 90° dorsal emergence, and -1 indicates 90° ventral emergence. Statistical comparisons across genotypes demonstrate that axons in both *unc-6* and *slt-1* mutants are misoriented relative to wild-type axons. These comparisons can be found in Figures S1B, 3C, S2B, S3C.

Second, I have concerns that the analysis of unc40 polarization may be misleading in some cases when there appears to indeed be accumulation in the growth cone, but since the only analysis shown is relative to the rest of the cell, that can be lost.

Thank you for sharing your concerns about the UNC-40 polarization quantifications. We have separately compared the value of the integrated density of UNC-40::GFP in each cellular domain (vSLNC-contacting area and the dorsal soma) between genotypes. While we did not include these comparisons in the original manuscript, we have now included them in the revised manuscript. Overall, these data support our conclusions that UNC-40 mispolarization occurs across the entire cell (Fig. S1F,G; S2E-H; S3E,F).